

# Quench dynamics of entanglement from crosscap states

Konstantinos Chalas[1], Pasquale Calabrese[1,2] and Colin Rylands[1]

**1** SISSA and INFN Sezione di Trieste, via Bonomea 265, 34136 Trieste, Italy
**2** International Centre for Theoretical Physics (ICTP), Strada Costiera 11, 34151 Trieste, Italy

## Abstract

The linear growth of entanglement after a quench from a state with short-range correlations is a universal feature of many body dynamics. It has been shown to occur in integrable and chaotic systems undergoing either Hamiltonian, Floquet or circuit dynamics and has also been observed in experiments. The entanglement dynamics emerging from long-range correlated states is far less studied, although no less viable using modern quantum simulation experiments. In this work, we investigate the dynamics of the bipartite entanglement entropy and mutual information from initial states which have long-range entanglement with correlation between antipodal points of a finite and periodic system. Starting from these *crosscap* states, we study both brickwork quantum circuits and Hamiltonian dynamics and find distinct patterns of behaviour depending on the type of dynamics and whether the system is integrable or chaotic. Specifically, we study both dual unitary and random unitary quantum circuits as well as free and interacting fermion Hamiltonians. For integrable systems, we find that after a time delay the entanglement experiences a linear in time decrease followed by a series of revivals, while, in contrast, chaotic systems exhibit constant entanglement entropy. On the other hand, both types of systems experience an immediate linear decrease of the mutual information in time. In chaotic systems this then vanishes, whereas integrable systems instead experience a series of revivals. We show how the quasiparticle and membrane pictures of entanglement dynamics can be modified to describe this behaviour, and derive explicitly the quasiparticle picture in the case of free fermion models which we then extend to all integrable systems.

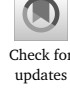
doi:[10.21468/SciPostPhys.19.5.132](10.21468/SciPostPhys.19.5.132)

# 1  Introduction

The standard protocol for studying the non-equilibrium dynamics of closed quantum systems is the quantum quench [1]. The system is prepared in some initial state and allowed to undergo unitary dynamics which is then probed using an appropriate diagnostic. Typically, the initial state is chosen to be of simple form and with low entanglement, either a product state or the ground state of a specific initial Hamiltonian. The reasons for this are both conceptual and practical. On the practical side, the simplistic nature of the initial states can allow for exact treatment in specific cases and a more straightforward analysis of the results, forgoing the need to disentangle complicated initial state correlations from non-equilibrium effects. In addition, the presence of entanglement in quantum systems presents a major challenge to numerical approaches based on matrix product algorithms. Therefore, by initiating the system in a lowly entangled state, numerical techniques can be used to track the system evolution, at least up until the entanglement barrier. On the conceptual side, these states present the opportunity to understand how the build up of correlations and entanglement between different spatial regions occurs as the system evolves and how it is eventually brought to local equilibrium [2,3]. Moreover, simple product states can routinely be engineered in an array of experimental platforms and so theoretical results can be confronted with experimental data. A vast amount of studies have been performed within this setting leading to a rapid advancement of the field of non-equilibrium quantum matter [4–9]. To maintain this level of growth, however, it is necessary to go beyond this paradigm and obtain new techniques and insights in complementary settings. In this paper, we take a small step in this direction and study quench dynamics from translationally invariant initial states which exhibit long-range entanglement. In particular, we study the dynamics of the entanglement entropy and mutual information in quenches to integrable and chaotic models undergoing unitary Hamiltonian and brickwork circuit evolution. We use exact analytic methods backed up by numerical simulation and show how the dynamics emerging from these states can be described within the existing set of effective theories upon suitable modification.

    The bipartite entanglement entropy between a subsystem and its complement is one of the most instructive quantities that one can calculate in a quantum many body system [10]. It can reveal universal information regarding the nature of the non-equilibrium state and its

approach to local equilibrium. For integrable or close to integrable models, quenched from lowly entangled states, the quasiparticle picture elegantly captures the leading order dynamics of the entanglement entropy [11]. Using minimal input about the initial state correlations it can provide quantitative predictions as well as physical insight into the underlying mechanisms of relaxation. Originally formulated in conformal field theory, it has since been extended to encompass generic integrable models [12, 13], include the effects of dissipation [14], or excited initial states [15, 16] and also predict the behaviour of a multitude of other quantities [17–32]. The picture relies on the existence of a set of long lived quasiparticles which are produced locally by the quench in correlated pairs, or multiplets, and then propagate ballistically throughout the system. As the members of the pairs separate and enter different spatial regions they carry with them correlations inherited from the initial state thereby creating entanglement. Given the initial correlations of the quasiparticle pairs and their kinematic data, the evolution of the entanglement entropy can be found.

In the absence of a stable set of quasiparticle excitations, such as in chaotic systems for instance, an alternative approach known as the entanglement membrane picture can be used to study the dynamics of the entanglement entropy [33–35]. This effective theory was first put forth in the context of random unitary quantum circuits but has also been extended to accommodate other systems as well. Therein, the calculation of Rényi entropies was recast as the computation of the partition function of an effective classical spin model. Within this picture the spins coincide with different possible pairings between the different replicas and the entropy is obtained via the free energy of a domain wall between different pairings. Armed with the energy density of the domain wall, the entanglement entropy can be readily calculated, however this can be difficult to determine ab initio [35–37].

**Initial states:** We consider a system of $2L$ qudits with local Hilbert space dimension $q$, located at positions denoted by $x$ with periodic boundary conditions, $x = x + 2L$. The system is initialized in the state $|\mathcal{M}\rangle$ defined as

$$|\mathcal{M}\rangle = q^{-L/2} \bigotimes_{x=1}^{L} \sum_{i,j=0}^{q-1} \mathcal{M}_{ij} |i\rangle_x \otimes |j\rangle_{x+L} \,. \tag{1}$$

Here, $\mathcal{M}$ is a $q \times q$ unitary matrix which specifies the initial state and $|i\rangle_z$, $j = 0, \ldots, q-1$ are a set of basis states for $\mathbb{C}^q$ on site $z$. It is normalized so that $\langle \mathcal{M}|\mathcal{M}\rangle = (\text{tr}[\mathcal{M}\mathcal{M}^\dagger])^L/q^L = 1$. Note that in conventional cases the initial state couples together adjacent sites, but in this work we instead couple together sites which are diametrically opposite and thus create long-range entanglement in the initial state. The state can be viewed as being made up of generalized Einstein-Podolsky-Rosen (EPR) pairs which are placed so that they are maximally distant from each other. States of this form also go under the name of entangled antipodal pair states [38, 39].

A simple example of these states is the crosscap state, see Figure 1, defined for $q = 2$ to be

$$|\mathcal{C}\rangle = 2^{-L/2} \bigotimes_{x=1}^{L} \sum_{i=0}^{1} |i\rangle_x \otimes |i\rangle_{x+L} \,. \tag{2}$$

States of this type were introduced as a spin chain realization of a crosscap state in quantum field theory [40–43]. The name arises as they are associated to boundary conditions on cylinder partition functions which have a crosscap topology i.e. antipodal points at the end of the cylinder are identified. These boundary conditions were studied in conformal field theories and then integrable field theories [43–48]. In the latter context they were identified

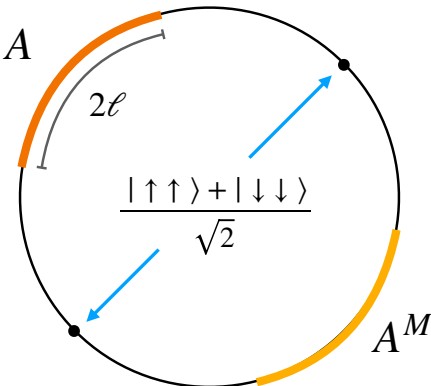

Figure 1: Illustration of the initial state and subsystems of interest. We consider a qudit chain of length $2L$, in which qudits on opposite sides of the system are initially entangled. For the crosscap state, $|\mathcal{C}\rangle$, antipodal points are prepared in an EPR state given by $\frac{|\uparrow\rangle_x|\uparrow\rangle_{x+L}+|\downarrow\rangle_x|\downarrow\rangle_{x+L}}{\sqrt{2}}$. The subsystem, $A$, is taken to be a contiguous block of $2\ell$ qudits. These are initially maximalliy entangled with the qudits in $A^M$ which is diametrically opposite $A$.

as integrable boundary conditions allowing for exact computation of their boundary partition functions [49, 50]. For quantum quenches to integrable models, it is known that integrable boundary conditions can be associated with a special set of integrable initial states for which the quench dynamics can be studied exactly [51]. The state $|\mathcal{C}\rangle$ was identified as such a state for the integrable spin chains and its overlaps with Bethe eigenstates were computed exactly [52–54].

We are interested in the properties within a subsystem $A$ which we take to be formed of $2\ell$ contiguous spins $A = [2, 2\ell + 1]$ and, for the most part, we also take $\ell < L/2$. Despite possessing long-range entanglement, the reduced density matrix of $A$, for our chosen initial states can be computed exactly and is given by,

$$
\begin{aligned}
\rho_A &= \text{tr}_{\bar{A}}[|\mathcal{M}\rangle\langle\mathcal{M}|] \\
&= \frac{1}{q^{2\ell}} \bigotimes_{x\in A} \sum_{i,j,k} \mathcal{M}_{i,j}\mathcal{M}_{k,j}^* |i\rangle_x \langle k|_x \qquad (3) \\
&= \frac{1}{q^{2\ell}} \mathbb{1}, \qquad (4)
\end{aligned}
$$

where $\bar{A} = [2\ell + 2, 2L + 1]$ is the complement of $A$ and in going to the last line we have used the unitarity of the matrix $\mathcal{M}$. Thus, the initial state appears locally indistinguishable from the infinite temperature state, as long as $\ell \leq L/2$. This is in sharp contrast to more typical quench setups were one might associate this to the endpoint of evolution. For instance, a system undergoing random unitary dynamics locally approaches the infinite temperature state in the long time limit. It should be noted, however, that for the dynamics we will consider, the states $|\mathcal{M}\rangle$ are not stationary even though they might appear as such, at least locally, in some cases.

We shall also be interested in the fate of the long-range correlations which exist in the initial state. With this in mind, we can instead take the subsystem to be composed of both $A$ and its mirror image across the chain, $A^M = [L + 2, L + 2\ell + 1]$. In this case, the reduced state

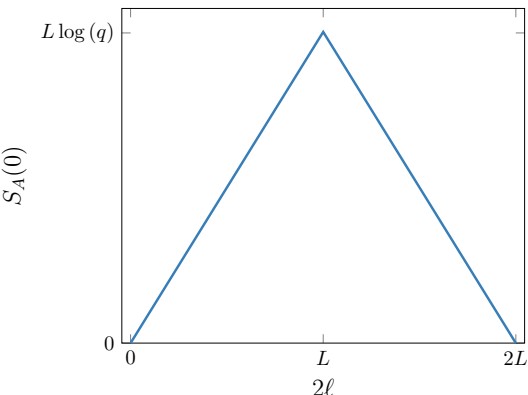

Figure 2: The bipartite entanglement entropy of a subsystem $A$ of length $2\ell$ in a system of total length $2L$, of the initial state $|\mathcal{M}\rangle$, as a function of the subsystem length $2\ell$.

is a pure state and the reduced density matrix is given by

$$\rho_{A \cup A^M} = |\mathcal{M}_A\rangle \langle \mathcal{M}_A| \,, \tag{5}$$

where $|\mathcal{M}_A\rangle$ is given by (1) but with the tensor product restricted to $x \in A$.

**Entanglement entropy:** Our primary quantity of interest will be the bipartite entanglement entropy between $A$ and $\bar{A}$. The Rényi entanglement entropy of $A$ is defined as

$$S_A^{(n)}(t) = \frac{1}{1-n} \log \mathrm{tr}\big[\rho_A^n(t)\big], \quad n \in \mathbb{R}, \tag{6}$$

where $\rho_A(t)$ is the reduced density matrix at time $t$ after the quench. From this the von Neumann entropy is obtained in the replica limit, $S_A(t) = \lim_{n \to 1} S_A^{(n)}(t)$. Using the form of the reduced density matrix we have that initially

$$S_A^{(n)}(0) = 2 \min[\ell, L-\ell] \log(q)\,, \tag{7}$$

where we have allowed for $\ell > L/2$. Hence, these states have maximal bipartite entanglement with a scaling given by the Page curve [55], see Figure 2. Thus, our set up could be considered opposite to the typical scenario where the initial entanglement is minimal. We are also interested in the Rényi mutual information between the subsystem $A$ and its mirror $A^M$. This is defined as

$$I_{A:A^M}^{(n)}(t) = S_A^{(n)}(t) + S_{A^M}^{(n)}(t) - S_{A \cup A^M}^{(n)}(t)\,. \tag{8}$$

We can use this to probe how the long-range, initial state correlations evolve in time. In the initial state we can use both (4) and (5) to find,

$$I_{A:A^M}^{(n)}(0) = 4 \min[\ell, L-\ell] \log(q)\,, \tag{9}$$

which is again maximal. Note that while the entanglement entropy of the contiguous subsystem is the same as the long time limit of random dynamics, the fact that the initial state is not stationary is betrayed by the mutual information. One can expect that the stationary state generated by a local Hamiltonian system or quantum circuit will have vanishing mutual

information. Thus, in the case of chaotic systems, $I_{A:A^M}(t)$ will serve as a useful probe of the dynamics. Moreover, in typical quench protocols the quasiparticle and membrane pictures provide two qualitatively different predictions for the behaviour of $I_{A:A^M}(t)$, as opposed to $S_A^{(n)}(t)$ for which both predict linear in $t$ growth and eventual saturation [56]. In quenches from lowly entangled states, the mutual information can be used to expose the different qualitative entanglement dynamics of chaotic and integrable systems [57–59], although this can also be done using a single interval [60, 61]. Thus, it is a natural tool for us to employ in the current scenario as well.

In this paper we will study the quench dynamics of $S_A^{(n)}(t)$ and $I_{A:A^M}^{(n)}(t)$ from the states $|\mathcal{M}\rangle$, in two broad classes of models: brickwork quantum circuits and Hamiltonian dynamics. Within each class of system we shall consider several different examples. For quantum circuits, we will examine dual unitary circuits, including the special case of swap gates, and random unitary circuits. In the second class of systems, we study two types of integrable models, the free fermion tight binding chain and its interacting generalization which is equivalent to the XXZ model. In all cases we find analytic expressions for $S_A^{(n)}(t)$ and $I_{A:A^M}^{(n)}(t)$ whose behaviour is then interpreted using the quasiparticle and entanglement membrane pictures after suitable modification. The entanglement dynamics of states with long-range entanglement similar to our states have recently been examined using the entanglement link representation [62, 63] whose connection with the quasiparticle picture has been noted [64].

The remainder of the paper is structured as follows. In section 2 we discuss the entanglement dynamics in brickwork circuits. First, we introduce the system and its corresponding graphical notation, after which we study the simple case of a circuit built from swap gates. This is done through a mix of exact calculations and the quasiparticle picture. Following this, the cases of random unitary and dual unitary gates are considered. The entanglement entropy and mutual information are obtained using approximate analytic methods and the results are compared to the entanglement membrane picture. In section 3 we investigate the entanglement evolution under Hamiltonian dynamics. We start by considering the case of a free fermion model whose dynamics are determined exactly. The quasiparticle picture for the system is obtained and compared to exact numerics. After this we consider the interacting, yet integrable model using the quasiparticle picture. Some details of these calculations are found in appendices A and B. In section 4 we conclude and comment on future directions.

## 2 Quantum circuits

We begin our investigation of the quench dynamics from long-range entangled states by examining the case of brickwork quantum circuits. In these models, the dynamics is generated by integer applications of the time evolution operator, $\mathbb{U}$, such that

$$|\mathcal{M}(t+1)\rangle = \mathbb{U}(t+1)|\mathcal{M}(t)\rangle, \quad t \in \mathbb{N}_0, \tag{10}$$

where $|\mathcal{M}(0)\rangle = |\mathcal{M}\rangle$ and

$$\mathbb{U}(t) = \bigotimes_{x \text{ odd}} U_{x,x+1}(x,t) \bigotimes_{x' \text{ even}} U_{x',x'+1}(x',t). \tag{11}$$

Here, $U_{x,x+1}(x,t)$ are unitary operators acting on the whole chain, but non-trivially only at sites $x, x+1$ as the unitary $q^2 \times q^2$ matrix $U(x,t)$. The latter are known as local gates and they can be conveniently represented using the standard diagrammatic notation of tensor networks [65]. In this notation, we associate to each local Hilbert space a line, or leg, and the

action of the local gate, or its conjugate, is represented by a colored box,

$$U(x,t) = \text{█} , \qquad U^\dagger(x,t) = \text{█} . \tag{12}$$

Within this framework, matrix multiplication is represented by connecting lines so that, for example, the unitarity of the local gates is represented as

$$\mathbb{1} = U^\dagger(x,t)U(x,t) = \text{█} = |\ | , \tag{13}$$

with the vertical straight lines denoting $\mathbb{1}$. Using this, one may depict the time evolution operator and its conjugate as

$$\mathbb{U}(t) = \text{█} , \tag{14}$$

$$\mathbb{U}^\dagger(t) = \text{█} , \tag{15}$$

which we have illustrated explicitly for $L = 8$ and from which we can see where the name brickwork circuit derives. The above diagrams should be viewed with periodic boundary conditions which connect the left most leg in the middle with the rightmost middle leg, leaving $2L$ uncontracted legs on the top and bottom. We can also depict our initial state in this manner via

$$|\mathcal{M}\rangle = \bigotimes_{x=1}^{L} {}^{x} \smile {}^{x+L} , \ |\mathcal{M}^\star\rangle = \bigotimes_{x=1}^{L} {}^{x} \smile {}^{x+L} , \tag{16}$$

where the indices above the legs indicate which site they are associated to. Combining these elements, the time evolved state is represented as

$$|\mathcal{M}(t)\rangle = \text{█} , \tag{17}$$

which we have depicted explicitly for $L = 8$ and $t = 3$. As before, periodic boundary conditions should be assumed which contract open legs on the left and right, leaving only $2L$ uncontracted legs at the top. The advantage of this notation is not just that it provides an illustrative graphical depiction of our quantities of interest, but also that, given a simple set of rules, it allows for calculations to be performed in an efficient and accessible manner. In this aspect, they are much the same as Feynman diagrams.

To introduce the rules associated to these diagrams and, moreover, in order to compute the entanglement entropies, we must work with multiple replicas of the theory. To simplify the notation it is then useful to introduce the replicated, folded, gate and initial state. They are depicted as

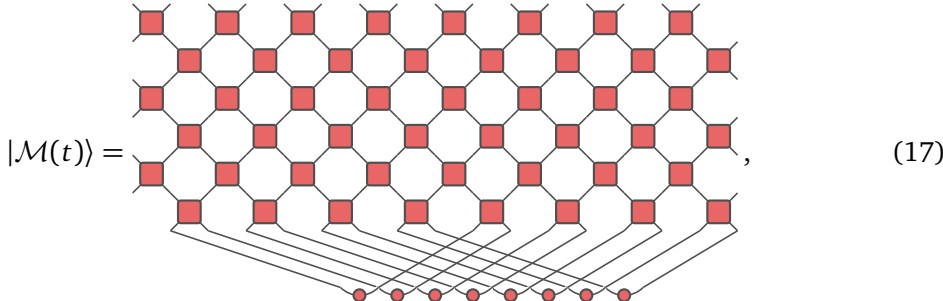

$$[U(x,t) \otimes U^\star(x,t)]^{\otimes_r n} = \text{█} \equiv \boxed{n} , \tag{18}$$

$$[|\mathcal{M}\rangle \otimes |\mathcal{M}^\star\rangle]^{\otimes_r n} = \bigotimes_{x=1}^{L} {}^{x} \smile {}^{x+L} \equiv \bigotimes_{x=1}^{L} {}^{x} \smile {}^{x+L} , \tag{19}$$

where $\otimes_r$ denotes the tensor product over replicas. On this folded and replicated space we also introduce two special states. The first, which is denoted by an empty circle, couples together the forward and backward time evolution on the same replica,

$$\left[\sum_{i=1}^{q} |i\rangle_x \otimes |i\rangle_x\right]^{\otimes_r n} \equiv \varphi.$$
(20)

Whereas the second, which we denote with an empty square, instead couples the forward time evolution on one copy to the backwards time evolution of the previous copy,

$$\sum_{i_1,\ldots,i_n=1}^{q} \left[|i_1\rangle_x \otimes |i_2\rangle_x\right] \otimes_r \left[|i_2\rangle_x \otimes |i_3\rangle_x\right] \otimes_r \cdots \otimes_r \left[|i_n\rangle_x \otimes |i_1\rangle_x\right] \equiv \square.$$
(21)

Using these states we introduce the basic rule of these diagrams, which derive from the unitarity of the local gate. These are

$$\stackrel{\circ}{\boxed{n}}\stackrel{\circ}{=}\varphi\varphi, \qquad \stackrel{\square}{\boxed{n}}\stackrel{\square}{=}\varphi\varphi, \qquad \stackrel{}{\boxed{n}}\stackrel{}{=}\downarrow\downarrow, \qquad \stackrel{}{\boxed{n}}\stackrel{}{=}\downarrow\downarrow.$$
(22)

In addition, the unitarity of the matrix $\mathcal{M}$ which defines the initial state provides us with the following relations,

$$\overset{x \quad x+L}{\underset{}{\circ}} = \frac{1}{q^n} \overset{x+L}{\circ} \,, \qquad\qquad \overset{x \quad x+L}{\underset{}{\circ}} = \frac{1}{q^n} \overset{x}{\circ},$$
(23)

$$\overset{x \quad x+L}{\underset{}{\square}} = \frac{1}{q^n} \overset{x+L}{\square} \,, \qquad\qquad \overset{x \quad x+L}{\underset{}{\square}} = \frac{1}{q^n} \overset{x}{\square}.$$
(24)

In the context of local product states, an initial state with these properties is known as solvable [66]. Finally, we note the following identities,

$$\overset{\circ}{\underset{\circ}{\circ}} = \overset{\square}{\underset{\square}{\circ}} = q^n \,, \qquad \overset{\square}{\underset{\circ}{\circ}} = q \,,$$
(25)

which follow from the definition of the square and circles states (20),(21). Combining this set of notations, we can ultimately depict the trace which appears in (6) as

$$\text{tr}\left[\rho_A^n(t)\right] =$$

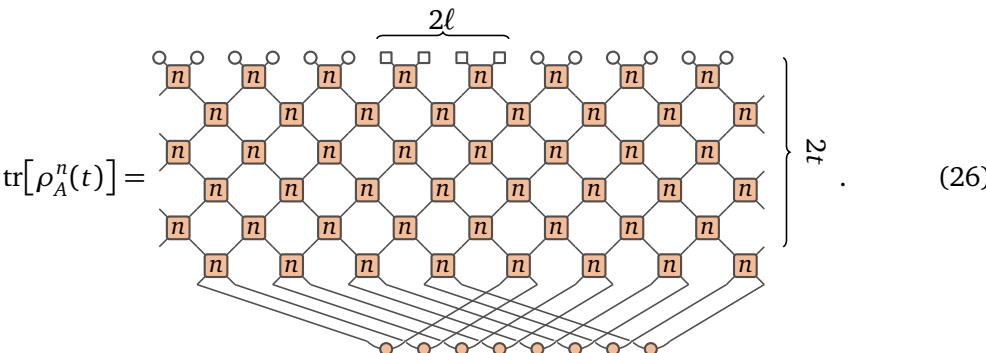

(26)

This diagram can be significantly reduced using the unitarity of the gates and the solvability of the initial state matrix. In particular, the square and circle states at the top of the diagram can be moved downward using the rules (22). When $4t - 2 + 2\ell \leq L$ and $L > 2\ell$ we obtain a

diagram of the form

$$\text{tr}\big[\rho_A^n(t)\big] = \frac{1}{q^{(4t-2+2\ell)n}} \qquad\qquad\qquad\qquad\qquad\qquad\qquad\qquad$$

$$= \frac{1}{q^{2(n-1)\ell}} \, .$$

(27)

Here, we have used the fact that when $t < L/4 - \ell/2$ no pairs of initially entangled sites appear in the backwards light cone of the subsystem $A$. As a result, one can use (23) to obtain the circle states on the bottom. From this, we see that unitarity of the system imposes that,

$$S_A^{(n)}(t) = 2\ell \log(q), \qquad t < \frac{L - 2\ell}{4}.$$

(28)

After this time, the entropy need not remain constant and we can again use unitarity to reduce the relevant diagram. For $4t - 2 + 2\ell = L + m$ and $m \le 2\ell$, we find that

$$\text{tr}\big[\rho_A^n(t)\big] = \begin{cases} \frac{\mathcal{J}_\text{e}(2t-1,m)}{q^{(n-1)(2\ell-m)+n(4t-2-m)}}, & L \text{ even,} \\ \frac{\mathcal{J}_\text{o}(2t-1,m)}{q^{(n-1)(2\ell-m-1)+n(4t-m-1)}}, & L \text{ odd,} \end{cases}$$

(29)

where the subscript e,o refer to the cases where $L$ is even or odd respectively in which case $m$ must also be even or odd. The newly introduced quantities $\mathcal{J}_\text{e,o}(p,m)$, are defined through the diagrammatic representations,

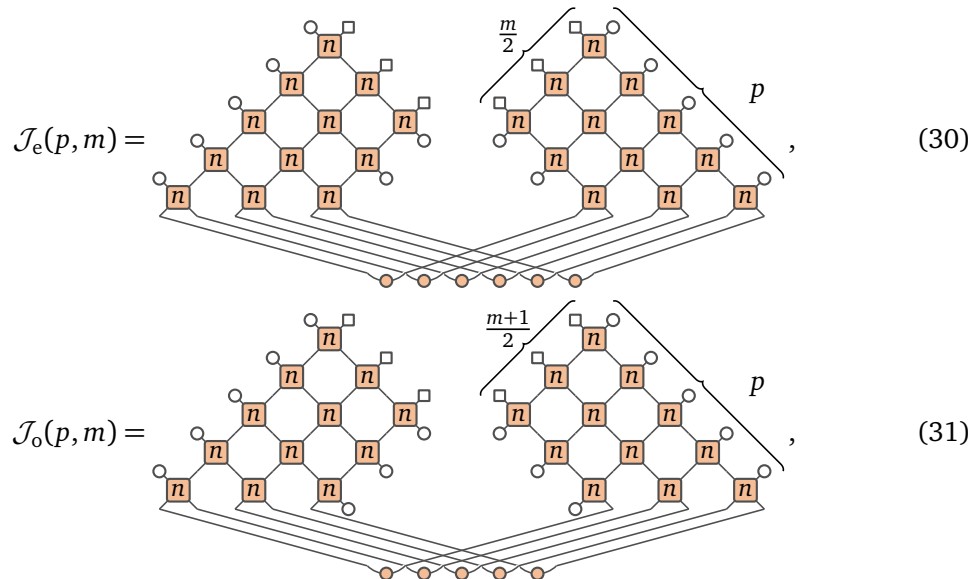

$$\mathcal{J}_\text{e}(p,m) = \qquad\qquad\qquad\qquad\qquad\qquad\qquad\qquad , \qquad (30)$$

$$\mathcal{J}_\text{o}(p,m) = \qquad\qquad\qquad\qquad\qquad\qquad\qquad\qquad , \qquad (31)$$

where we assume $p > m/2$. At this point the unitarity of the gates and initial state does not allow us to compute $\mathcal{J}_\text{e,o}$ and so we need to consider some specific examples.

A similar procedure can be carried out in the case where the subsystem is $A \cup A^M$. In that case, we obtain that for $t < \frac{L-2\ell}{4}$

$$\text{tr}\big[\rho_{A \cup A^M}^n(t)\big] = \begin{cases} \mathcal{D}_\text{e}(2t-1, 2\ell), & L \text{ even,} \\ \frac{1}{q^{2n}} \mathcal{D}_\text{o}(2t-1, 2\ell), & L \text{ odd,} \end{cases}$$

(32)

where again the subscripts e,o refer to the cases of even and odd $L$ respectively. The new quantities, $\mathcal{D}_{e,o}(2t-1, 2\ell)$, are once again defined through their diagrammatic representations to be

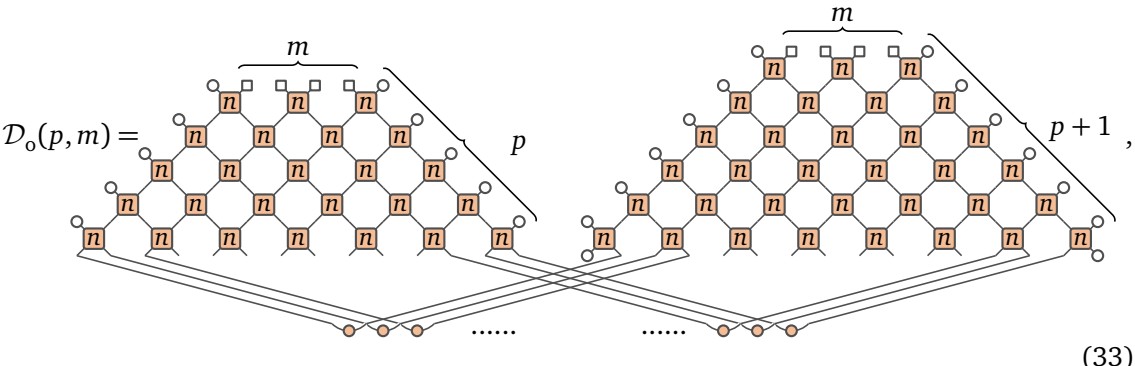

$$\tag{33}$$

and

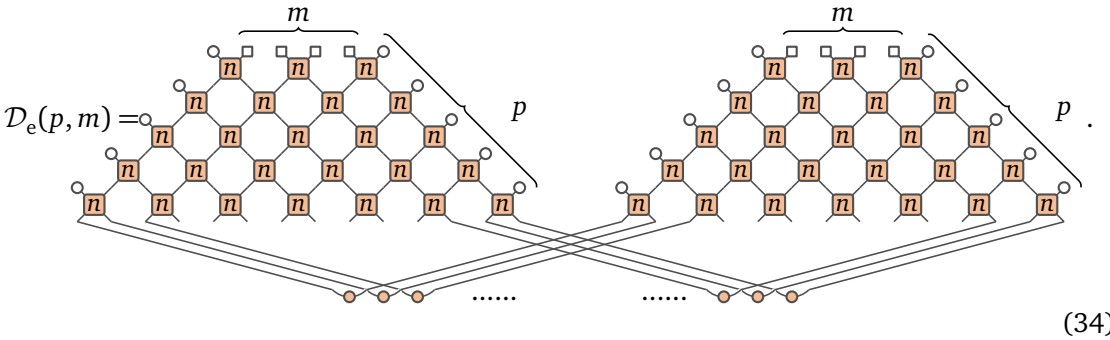

$$\tag{34}$$

These expressions can be further reduced using unitarity depending on the values of $p, m$, but it is more convenient to work with them in the form presented above. To proceed further, we now turn to the consideration of several specific examples.

## 2.1 Swap gate circuit

The first example we consider is the case where the local gates are given by $U(x, t) = P$, $\forall x, t$ where $P$ is the swap gate. It is defined by its action on two sites as,

$$P \left| i \right\rangle_x \otimes \left| j \right\rangle_{x+1} = \left| j \right\rangle_x \otimes \left| i \right\rangle_{x+1}, \tag{35}$$

i.e. that it swaps the states at the sites on which it acts. In terms of diagrams this is represented as

$$P = \times, \tag{36}$$

meaning that one should contract indices along the diagonal lines only. Using this form of gate, the quantities $\mathcal{J}_{e,o}(n, m)$ can be straightforwardly calculated giving,

$$\mathcal{J}_e(p, m) = q^{n(2p-m)-m(n-1)}, \tag{37}$$

$$\mathcal{J}_o(p, m) = q^{n(2p-m+1)}. \tag{38}$$

Upon inserting these into (29) and using (6), we find that, for $m_t \equiv 4t + 2\ell - 1 - L$ with $2\ell \leq m_t > 0$,

$$S_A^{(n)}(t) = \begin{cases} 2\ell \log(q), & L \text{ even,} \\ (2\ell - m_t) \log(q), & L \text{ odd.} \end{cases} \tag{39}$$

Thus, while for even $L$ the entanglement entropy remains constant, this is not the case for odd $L$ which experiences a linear in $t$ decrease for $t > \frac{L-2\ell}{4}$. This linear decrease terminates at $t_{\min} = \lfloor \frac{L+1}{4} \rfloor$. At this time we arrive at a minimum value of entanglement, $S_A^{(n)}(t_{\min}) = 2\log(q)$ for $\frac{L+1}{2}$ being odd or equivalently $\ell$ being odd and $S_A^{(n)}(t_{\min}) = 0$ for $\frac{L+1}{2}$ or $\ell$ being even. The odd-even effect of (39) can be traced back to what sites are entangled by the initial state and how those correlations are propagated by the circuit. For the half system size, $L$, being even, the initially entangled pairs of qudits are both on even or both on odd sites. Furthermore, by inspecting the form of the gate in (36), we see that information from sites is chirally propagated, meaning that information on an even numbered site is propagated solely to the left in the swap circuit while information on odd sites is propagated to the right. Thus, when the initial state entangles only even sites or only odd sites but not even with odd, the entanglement remains constant, in effect the entanglement always remains between sites which are a distance $L$ apart. In contrast, when $L$ is odd, the initial state possesses correlations between even sites and odd sites. The information from these sites then propagates in opposite directions and so the distance between sites which are entangled decreases. Once a pair of entangled sites is within $A$, the $S_A^{(n)}(t)$ experiences a drop in entanglement of $2\log(q)$.

Naturally, the preceding discussion can be formulated in the language of the quasiparticle picture. The quench produces quasiparticles which propagate ballistically, with dimensionless velocity 2, throughout the system. In this case left moving particles are produced at the even sites and right movers are produced at the odd sites. If $L$ is even, then correlated pairs of quasiparticles have the same chirality and because of their constant velocity, always remain a distance $L$ apart. Therefore, they cannot both be inside the subsystem at the same time and change the entanglement entropy. In the case of odd $L$ however, the quasiparticle pairs have opposite chirality and so will enter the subsystem at a certain point in time. This happens first at $t = \frac{L-2\ell}{4}$ when the pair which are initially equidistant from the center of the subsystem enter inside it. This leads to a linear decrease in time of $S_A^{(n)}(t)$ with a slope of $4\log(q)$. This continues until $t = \lfloor \frac{L+1}{4} \rfloor$ time steps, at which point the state resembles a lowly entangled state, where the entangled quasiparticle pairs sit on adjacent sites. After this the system experiences a linear increase in entanglement until it returns to its original value. Using this, along with the fact that the dynamics should be periodic in time with period $\lfloor \frac{L+1}{2} \rfloor$ we can construct the dynamics at aribitrary times. The result is plotted in Figure 3 in which we see a sequence of decreases and increases in entanglement.

Another straightforward calculation allows us to examine the case where the susbsystem consists of $A$ and its mirror. Here we find that

$$\mathcal{D}_e(p,m) = 1\,, \tag{40}$$

$$\mathcal{D}_o(p,m) = \frac{1}{q^{n(\max[2m-2,4p+2])}}\,. \tag{41}$$

Inserting these in (32) we find that

$$I_{A:A^M}^{(n)}(t) = \begin{cases} 4\ell\log(q), & L \text{ even,} \\ 4(\ell - \min[\ell, 2t])\log(q), & L \text{ odd.} \end{cases} \tag{42}$$

Therefore, as might be expected, when $L$ is even there is no change in the mutual information while when $L$ is odd we have a linear in time decrease in the mutual information up to $t = \frac{\ell}{2}$ whereupon it stays zeros. After $t = \frac{L-2\ell}{4}$ one can show that $\mathrm{tr}[\rho_{A\cup A^M}^n(t)]$ starts to increase again, compensating for the decrease in $\mathrm{tr}[\rho_{A,A^M}^n(t)]$ thereby maintaining the positivity of $I_{A:A^M}^{(n)}(t)$. Upon using the quasiparticle picture we can extend the above result to arbitrary times which shows, as before, a series of periodic-in-time revivals. This is plotted in Figure 3.

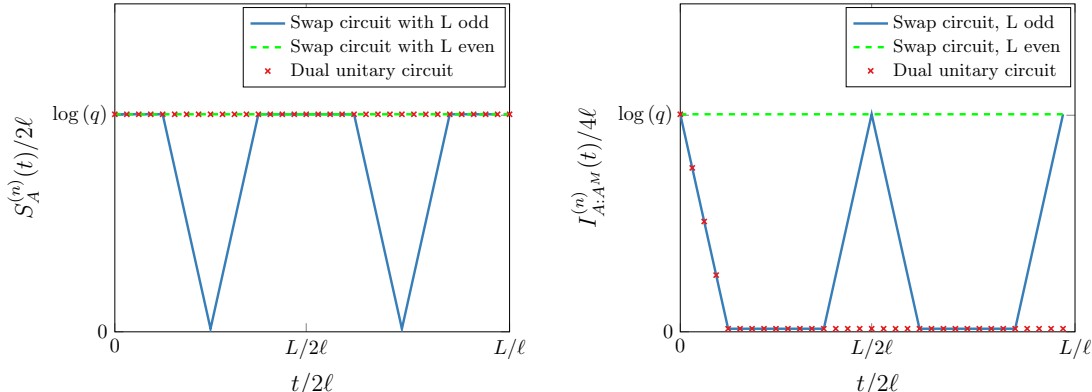

Figure 3: Left: $S_A^{(n)}(t)/2\ell$ as a function of $t/2\ell$ for the swap gate circuit with $L$ odd (solid blue line), the swap gate circuit with $L$ even (dashed green line) and generic dual unitary circuits as well (red symbols). Right: $I_A^{(n)}(t)/4\ell$ as a function of $t/2\ell$ for the same systems.

## 2.2 Random unitary circuit

Having seen how the entanglement dynamics behaves in the case of the swap circuit, which is integrable and noninteracting, we contrast this with the case of a random unitary circuit which is, instead, chaotic. In this case the local gates are independently drawn at random from the Haar ensemble and we shall be interested in quantities which are averaged over all possible realizations of the randomly chosen circuit.

Specializing to the case of $n = 2$ and averaging over all realizations, our diagrams obey an additional rule, namely

$$\boxed{2} = \alpha\left(\,\right), \qquad \boxed{2} = \alpha\left(\,\right), \tag{43}$$

where $\alpha = q/(q^2 + 1)$. These are standard results in the theory of Weingarten calculus and can be obtained by averaging the quantity $U \otimes U \otimes U^* \otimes U^*$ over all unitary matrices using the Haar measure [67, 68]. Using this new relation we can then reduce $\mathcal{J}_{e,o}(p, m)$ further. In particular, applying (43) to the top left and right most gates it is straightforward to show that

$$\mathcal{J}_{e,o}(p, m) = \alpha^2 q^2 \left[ \frac{1}{q^8} \mathcal{J}_{e,o}(p-1, m-4) + \frac{2}{q^4} \mathcal{J}_{e,o}(p-1, m-2) + \mathcal{J}_{e,o}(p-1, m) \right], \tag{44}$$

which is valid provided $p > m/2$ but must be supplemented by an appropriate set of boundary conditions. Similar recursion relations have been analysed in the context of random unitary dynamics from lowly entangled states yielding exact solutions [69]. In the present case, the boundary conditions are significantly more involved and we postpone a full analysis to future work, choosing instead to study them in the limit of large $q$. In that case it can be shown that the first term on the right hand side is the leading term. These terms lead to the largest number of contractions between states of the same type, i.e. $\,$ or $\,$. Other terms instead result in these being replaced with $\,$ diagrams meaning that they are subleading at large $q$. In that limit we find

$$\mathcal{J}_e(p, m) \simeq q^{4p-3m}, \tag{45}$$

$$\mathcal{J}_o(p, m) \simeq q^{4p-3m+1}. \tag{46}$$

These relations can be readily checked by inserting theses scaling forms into (44) and also by directly calculating $\mathcal{J}_{\mathrm{e,o}}(p,m)$ for $1 \leq m \leq 4$. Using these in (29) we then find that the annealed average entanglement entropy in the limit of large local Hilbert space dimension is

$$\langle S_A^{(2)}(t) \rangle_{\mathrm{a}} = 2\ell \log(q), \tag{47}$$

where $\langle \cdot \rangle_{\mathrm{a}}$ denotes the annealed average over realizations of the circuit. Thus in contrast to the swap gate we find that the entanglement entropy remains maximal and there is no even-odd effect.

This result can be interpreted using the entanglement membrane picture [33–35]. For random unitary circuits the local gate, after averaging over realizations, effectively projects the full local space onto a two dimensional subspace spanned by the non-orthogonal states (20) and (21). The representation of $\mathrm{tr}\left[\rho_A^2(t)\right]$ thus resembles a 2 dimensional lattice model with links taking on the values of the circle or square state. The line of states at the top imposes a specific boundary condition which creates two domain walls between the circle state on either side and square states in the middle. The entanglement entropy is then obtained by calculating the energy of these domain walls or membranes which start at the top and either terminate on the lower edge or on each other. In the standard scenario of lowly entangled initial states, there is no additional cost to terminating on the lower edge. In this scenario, however, since we have maximal entanglement across any single cut in the initial state there is a large energy cost for the membrane to terminate there. Thus, at any finite time, it is energetically favourable for the membrane to stretch horizontally across the subsystem rather than paying the cost associated to the lower boundary. We note that the original formulation of the membrane picture [33, 34] already allowed for the possibility of a highly entangled initial state but has typically only been applied to lowly entangled initial states. This leads to the expression (47). Therefore, from the point of view of the entanglement entropy, there appear to be no dynamics when quenching from $|\mathcal{M}\rangle$ to a random unitary circuit. Moreover, since we are performing an average over realizations, which has returned the maximum value, this should be true for all realizations.

To see that, indeed, there are non-trivial dynamics occurring we turn instead to the mutual information. To achieve this we must evaluate $\mathcal{D}_{\mathrm{e,o}}(p,m)$ which unfortunately does not admit such a nice recursion relation as $\mathcal{J}_{\mathrm{e,o}}$. However, we determine its scaling in the large $q$ limit as we did for the single interval. Again, the leading order term is the one which retains the most contractions between states of the same type. Within this approximation, when $2p < m$, we find that

$$\mathcal{D}_{\mathrm{e}}(p,m) \simeq \frac{1}{q^{4p+4}}, \tag{48}$$

$$\mathcal{D}_{\mathrm{o}}(p,m) \simeq \frac{1}{q^{4p}}. \tag{49}$$

From these, we can deduce that, at least for $t < \frac{L-2\ell}{4}$ and to leading order in large $q$,

$$\langle I_{A:A^M}^{(2)}(t) \rangle_{\mathrm{a}} \simeq 4(\ell - \min[\ell, 2t]) \log(q). \tag{50}$$

Thus, there is a linear decrease in the mutual information as was the case in the swap circuit for $L$ odd. Beyond $t > \frac{L-2\ell}{4}$ however, we do not expect similar behaviour in the chaotic and integrable dynamics as the latter experiences revivals while the former is not expected to, which we explain further below.

This result can also be interpreted using the membrane picture with only a slight modification. As we saw in the single interval case, the difference from the standard membrane

picture arises from the long-range correlated nature of the initial state. For the case in which the subsystem consists of both $A$ and $A^M$, we must properly account for the fact that these two disjoint intervals are entangled. As a result, the additional energetic penalty for the membrane to end on the lower boundary is exactly cancelled if there is a corresponding membrane also terminating on the initial state a distance $L$ away. With this modification, one can understand that the most favourable configuration is for the membranes emerging from all 4 entangling points to proceed vertically downwards at short time, leading to a linear decrease in the mutual information. After $t = \ell/2$ and provided $L > 4\ell$ the membranes are arranged horizontally across the disjoint parts of the subsystem leading to a vanishing of the mutual information. This prediction is in agreement with (50) but is applicable beyond $t > \ell/2$ indicating that the mutual information remains zero. This can be naturally encoded in the original formulation of the entanglement membrane picture [34] upon properly incorporating the spatial profile of the initial state entanglement. To properly derive this however one would need to obtain a solution to the recursion relations (44) beyond leading oreder in $1/q$

## 2.3 Dual unitary circuit

As a final example we examine a dual unitary circuit. These circuits have attracted much attention in recent years for their ability to provide insight into the dynamics of many body quantum systems, both chaotic and integrable while at the same time allowing for certain quantities to be computed exactly [61,66,69–83]. Their defining characteristic is that they are unitary when viewed in both the space and time directions. In terms of the folded represented gate, this property is depicted as

$$\text{(51)}$$

$$\text{(52)}$$

These relations are also satisfied by the swap gate, which is a special point in the space of dual unitary gates. Away from such special points, which also include the interacting but integrable, dual unitary XXZ circuit, dual unitary circuits are chaotic with properties which are distinct from random unitary gates. For a specific class of dual unitary gates, which includes all $q = 2$ gates it has been shown that, generically, the following relation holds [58],

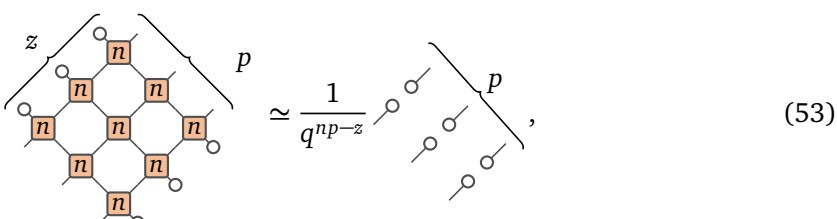

$$\text{(53)}$$

for large enough $z$ and arbitrary $p$. The essence of this relation is the leading eigenvector of the object on the left at $z = 1$ is a product over $\phi$ states. Thus, for large enough $z$, the left hand side becomes a projector onto this leading eigenvector, giving the expression on the right hand side. For special cases, such as the swap gate considered above, the leading eigenvector is not unique and this approach cannot be used. Using this, we can then reduce $\mathcal{J}_{e,o}$ to a form where it can be fully contracted provided $\frac{m}{2}$ is large enough. From this we find that as in the previous case, that the entanglement entropy remains constant

$$S_A^{(n)}(t) = 2\ell \log(q). \tag{54}$$

This result is valid for all Rényi indices, $n$. As above, this result is in agreement with the prediction of the appropriately modified membrane picture.

Lastly, we calculate the mutual information. For dual unitary circuits it is possible to reduce $\mathcal{D}_\mathrm{o}(p, m)$ completely using (51) because of the additional circle states on the bottom row of the right hand part of (33). Unfortunately, the same simplification cannot be applied to $\mathcal{D}_\mathrm{e}(p, m)$. Nevertheless, for $L$ being odd we can find that, at least for $t < \frac{L-2\ell}{4}$

$$I_{A:A^M}^{(n)}(t) = 4(\ell - \min[\ell, 2t])\log(q).\tag{55}$$

This result is valid for an arbitrary dual unitary circuit, including the swap gates discussed previously. In the case of the swap gates however, we can go beyond this time regime and show that revivals $I_{A:A^M}(t)$ can occur. For a generic dual unitary however, we expect no revivals to occur since dual unitary gates are generically chaotic. Once again the result (55) is in agreement with the prediction of the modified entanglement membrane picture presented for the random unitary circuit in the previous subsection. We plot this behaviour in Figure 3, comparing to the special case of swap gates.

## 3 Hamiltonian dynamics

In the previous section we have seen that a quantum circuit made of swap gates can exhibit nontrivial quench dynamics provided the half system size, $L$, is chosen to be odd. This manifests in a linear decrease of both the entanglement entropy and the mutual information, with the former only occurring after a time delay, which depends on both the total and subsystem sizes. In contrast, for the case of $L$ even, we saw that the time evolution was trivial and both quantities were constant. Both scenarios can be understood straightforwardly through the quasiparticle picture with the difference lying in how the initially correlated quasiparticle pairs are transported. To investigate these features further in an alternative setting we examine a system in which there is continuous time evolution generated by the Hamiltonian,

$$H_U = -\sum_{x=1}^{2L} c_x^\dagger c_{x+1} + c_{x+1}^\dagger c_x + 4U c_x^\dagger c_x c_{x+1}^\dagger c_{x+1}.\tag{56}$$

Here $c_x^\dagger$, $c_x$ are canonical fermionic creation and annihilation operators acting on site $x$, with the first two terms describing their hopping on the lattice and the last term describing their nearest neighbour density-density interaction which has a strength $U$. This model is integrable for all values of $U$ and is equivalent to the XXZ chain via Jordan-Wigner transformation [84–86]. The free fermion Hamiltonian at $U = 0$ is denoted by $H_0$. We choose to work with the fermionic version in order to avoid complications which can arise in the spin chain formulation when one is interested in disjoint subsystems [87]. Here, the local Hilbert space dimension is $q = 2$ and in the previously introduced notation, the local basis of states is defined so that $|0\rangle_x$ and $|1\rangle_x$ are the states which contain zero or one fermion,

$$c_x |0\rangle_x = 0, \qquad c_x^\dagger |0\rangle_x = |1\rangle_x.\tag{57}$$

To simplify matters, we shall restrict our attention to the quench dynamics emerging from the crosscap state (2). Starting from this crosscap state defined for the spin chain and performing a careful Jordan-Wigner transformation one finds that it is, in fact, a superposition of two fermionic Gaussian states, related to each other via a gauge transformation [48]. While the

entanglement dynamics of superpositions of Gaussian states can be studied analytically, it is much more involved [87]. To make progress, therefore, we shall study the fermionic crosscap state defined directly in terms of (57). Namely, we take

$$|\mathcal{C}\rangle = \frac{1}{2^{L/2}} e^{\sum_{x=1}^{L} c_x^\dagger c_{x+L}^\dagger} \bigotimes_{x=1}^{2L} |0\rangle_x \,, \tag{58}$$

which is equivalent to each of the individual Gaussian states in the spin chain realization. Moreover, we impose anti periodic boundary conditions, $c_{x+2L} = -c_x$ which ensure that the state is translationally invariant. As before, we will compute the dynamics of both $S_A(t)$ and $I_{A:A^M}(t)$ which can be done directly without the need for the replica trick.

When $U = 0$, the Hamiltonian reduces to the simple tight binding model of noninteracting fermions and in that case we can use the Gaussianity of the initial state (58) to compute $S_A(t)$ and $I_{A:A^M}(t)$ exactly [88]. This can then, in turn, be used to derive the quasiparticle picture for these quantities. In the interacting case, $U \neq 0$, it is no longer possible to perform an exact, first principles calculation of the entanglement entropy. It is now well established, however, that for the von Neumann entanglement entropy and mutual information, the quasiparticle picture can be applied to interacting integrable cases also [12, 19]. Rényi entropies, on the other hand, do not conform to the standard quasiparticle picture when there are interactions and it needs to be replaced using the methods of space-time duality [21,30,56]. For this reason we shall discuss only the von Neumann quantities. Our strategy will therefore be to first derive the quasiparticle picture in the noninteracting case, check it against exact numerics, and then adapt the results to the generic interacting model. As we explain in detail below, the emergent quasiparticle picture will necessarily be distinct from the standard quench due to presence of long-range correlations in the initial state.

## 3.1 Free fermion model

For a fermionic system in a Gaussian state, evolving according to the quadratic Hamiltonian, $H_0$, the entanglement entropy can be exactly computed via its two-point correlation function [88]. For this, we introduce the correlation matrix $C_A(t)$ within the subsystem $A$, whose matrix elements are

$$[C_A(t)]_{x,y} = \langle \mathcal{C}(t)| \mathbf{c}_x^\dagger \mathbf{c}_y |\mathcal{C}(t)\rangle \,, \quad x, y \in A, \tag{59}$$

where $|\mathcal{C}(t)\rangle = e^{-iH_0 t} |\mathcal{C}\rangle$ and $\mathbf{c}_x = (c_x, c_x^\dagger)$. Denoting the eigenvalues of this $4\ell \times 4\ell$ matrix by $\gamma_\alpha(t)$, $\alpha = 1, \dots, 4\ell$, the entanglement entropy is given by [88]

$$S_A(t) = -\frac{1}{2} \sum_{\alpha=1}^{4\ell} (\gamma_\alpha(t) \log[\gamma_\alpha(t)] + [1 - \gamma_\alpha(t)] \log[1 - \gamma_\alpha(t)]) \,. \tag{60}$$

The same procedure can also be used to obtain $S_{A \cup A^M}(t)$ by extending (59) to include all sites in the subsystem, i.e. $x, y \in A \cup A^M$. After introducing the Fourier transforms

$$c_x = \frac{1}{\sqrt{2L}} \sum_{n=1}^{2L} c_{k_n} e^{ixk_n} \,, \qquad c_{k_n} = \frac{1}{\sqrt{2L}} \sum_{x=1}^{2L} c_k e^{-ixk_n} \,, \tag{61}$$

with $k_n = \pi(n + 1/2)/L$ and using the fact that $e^{-iH_0 t} c_{k_n}^\dagger e^{iH_0 t} = e^{-it\epsilon(k_n)} c_{k_n}^\dagger$ where $\epsilon(k_n) = -2\cos(k_n)$ is the quasiparticle energy, we find that

$$[C_A(t)]_{x,y} = \frac{1}{4L} \sum_{n=1}^{2L} e^{ik_n(x-y)} \begin{pmatrix} 1 & e^{ik_n L - 2it\epsilon(k_n)} \\ e^{-ik_n L + 2it\epsilon(k_n)} & 1 \end{pmatrix} \,. \tag{62}$$

Here, we can note that the off-diagonal elements of $C_A(t)$ have a dependence on the size of the full chain $2L$, which serve as the imprint of the long-range entangled structure of $|\mathcal{C}\rangle$ across $L$ sites. Moreover, prior to the quench, since we restrict to $\ell < L/2$, the correlation matrix, $C_A(0)$ is diagonal with all equal eigenvalues $\gamma_\alpha(0) = 1/2$. This is in accordance with the fact that the initial state is maximally entangled. Furthermore, the above matrix has a Toeplitz structure, since the elements of this matrix only depend on the difference of their indices. This can be used to efficiently compute $C_A(t)$ numerically and after diagonalizing it, to use the resulting spectrum to obtain $S_A(t)$ exactly.

## 3.2  Quasiparticle picture

To start unravelling the quench dynamics of the crosscap state under $H_0$, it is instructive to first study the dynamics of the particle number, $N = \sum_x c_x^\dagger c_x$ restricted within the subsystem, i.e.

$$N_A(t) = \sum_{x \in A} c_x^\dagger(t) c_x(t). \tag{63}$$

Since $H_0$ conserves particle number globally i.e. $[N, H_0] = 0$, the expectation value inside the subsystem remains constant, $\langle N_A(t) \rangle = \ell$. However, higher order moments still exhibit nontrivial evolution. This can be probed through the variance, $\sigma_A^2(t)$, of the conserved charge inside $A$ defined as

$$\sigma_A^2(t) = \langle N_A(t) - \langle N_A \rangle \rangle^2 = \langle N_A^2(t) \rangle - \ell^2, \tag{64}$$

which can be calculated in one of two ways. First, we can proceed directly by using the definition (63) and employing the results of (62), obtaining

$$\sigma_A^2(t) = \sum_{x,y \in A} \langle c_x^\dagger(t) c_x(t) c_y^\dagger(t) c_y(t) \rangle - \ell^2. \tag{65}$$

Alternatively, we can relate it to the eigenvalues of the reduced correlation matrix $C_A(t)$ via [22, 89],

$$\sigma_A^2(t) = \frac{1}{2} \sum_{\alpha=1}^{4\ell} \gamma_\alpha(t) [1 - \gamma_\alpha(t)]. \tag{66}$$

Therefore, given an analytic expression for the charge fluctuations which can be straightforwardly calculated using (65), we can determine the time dependence of the eigenvalues of the correlation matrix $\gamma_\alpha(t)$ and, as a result, the entanglement entropy.

Switching to Fourier space and taking the thermodynamic limit,

$$\frac{1}{2L} \sum_{n=1}^{2L} f(k_n) \to \int_{-\pi}^{\pi} \frac{dk}{2\pi} f(k),$$

the variance of charge inside $A$ is given by [90],

$$\sigma_A^2(t) = \frac{\ell}{2} - \frac{1}{2} \sum_{x,y=1}^{2\ell} \int \frac{dk\,dq}{(2\pi)^2} e^{2it(\epsilon(q)-\epsilon(k))+i(q-k)(x-y-L)}, \tag{67}$$

where the first term can be recognized as the initial state value, which can be obtained from (66) using $\gamma_\alpha(0) = 1/2$. The second term constitutes the time dependent correction

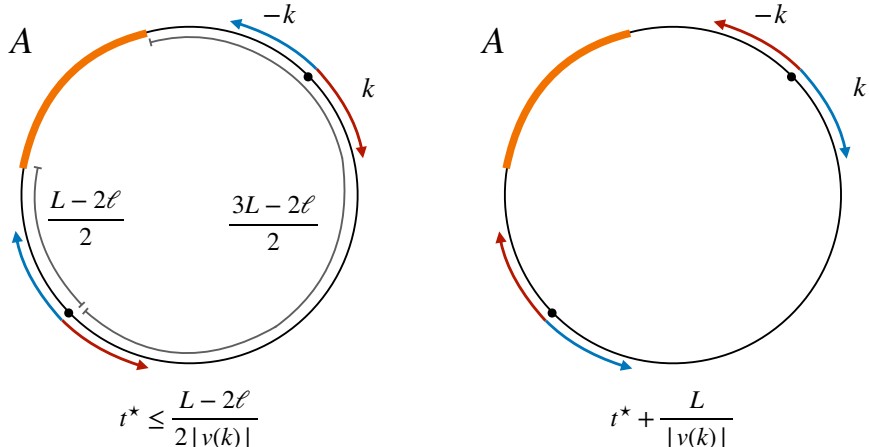

Figure 4: A depiction of the quasiparticle picture for the free fermion quench of the crosscap state, $|\mathcal{C}\rangle$. From each point in space, quasiparticles of momenta $k$ and $-k$ are emitted. These are depicted by the blue and red arrows. Quasiparticles are only correlated with the opposite momentum counterpart at the antipodal point i.e. blue arrows represent a correlated pair and likewise for red arrows. No correlations exist between quasiparticles represented by different colors. On the left we depict the situation for a time $t^\star \le \frac{L-2\ell}{4}$. A quasiparticle pair (blue arrows) emitted from points a distance $L/2 - \ell$ from $A$ both enter $A$ at the same time, at which point they reduce the entanglement between $A$ and $\bar{A}$. The other pair emitted from the same points (red arrows) must traverse the distance $3L/2 - \ell$ before both entering the subsystem. On the right we depict the situation for $t^\star + \frac{L}{|v(k)|}$. Here the red and blue arrows have swapped places but the otherwise the state is the same, leading to the use of $\tau_k$ in (68).

to the initial value and it can be evaluated using the multi-dimensional stationary phase approximation [17, 91]. Relegating the details to Appendix A, we find that

$$\sigma_A^2(t) = \frac{\ell}{2} - \int_{-\pi}^{\pi} \frac{dk}{2\pi} \max\left(0, 2\ell - |2\tau_k v(k) - L|\right), \tag{68}$$

$$\tau_k \equiv t \bmod \frac{L}{|v(k)|}, \tag{69}$$

where $v(k) = 2\sin(k)$ is the quasiparticle velocity. From the above result we will be able to understand the underlying quasiparticle dynamics and, accordingly, the entanglement entropy, so let us take some time to remark upon its form.

To begin, we should recall how the standard quasiparticle picture plays out. Therein, the quench *locally* produces pairs of *locally* correlated quasiparticles such that the quasiparticle with momentum $k$ produced at a point $x$ is correlated with the quasiparticle of momentum $-k$ also produced at $x$, and no others. A physical quantity is then computed by first knowing how each pair contributes to the quantity in question and then counting up the total contribution of all pairs. For $t \le 2L/v(k)$, the number of pairs indexed by $k$ inside the subsystem is given by the counting function $\max[0, 2\ell - 2|v(k)|t]$. Combining this with the contribution of each pair and integrating over the momenta we obtain the quasiparticle picture result. To extend this result into the regime where $t > 2L/|v(k)|$, we should make the replacement $t \to t \bmod 2L/|v(k)|$

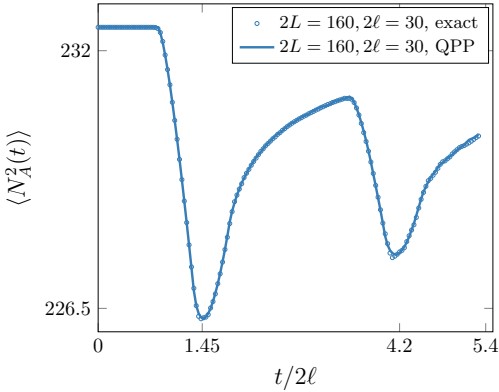
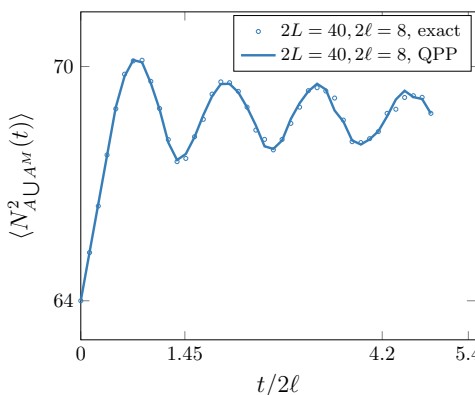

Figure 5: Left: $\langle N_A^2(t)\rangle = \ell^2 + \sigma_A^2(t)$ for total system size $2L = 160$ and subsystem size $2\ell = 30$, with the symbols being the exact calculation and the blue curve using the quasiparticle picture obtained from (68). Right: $\langle N_{A\cup A^M}^2(t)\rangle$, for total system size $2L = 40$ and subsystem size $2\ell = 8$. The blue curve is given by (71) and the symbols are the exact numerical result.

which merely keeps track of the fact that the quasiparticles return to their original position after a time $2L/|v(k)|$ [92]. Inspecting (68), we see that the integrand of the second term is reminiscent of the counting function with two modifications: the shift by $L$ and the use of the periodic time variable $\tau_k$. To understand the effect of these modifications we can use the intuition garnered from the swap gate circuit of the previous section.

Just like the standard case, quenching from the crosscap state also locally produces pairs of quasiparticles of opposite momentum at each point. In contrast, however, since the initial state only has correlation between sites which are a distance $L$ apart, the quasiparticle of momentum $k$ produced at $x$ is correlated with the quasiparticle of momentum $-k$ produced at $x + L$ and no others. Likewise, the quasiparticle of momentum $-k$ produced at $x$ is correlated only with the quasiparticle of momentum $k$ produced at $x + L$, see Figure 4. The counting function, $\max(0, 2\ell - |2tv(k) - L|)$ counts the number of non-locally correlated quasiparticle pairs indexed by $k$ and produced at $x$ and $x + L$ which are inside the subsystem at time $t$. We can check that this counting function vanishes for $t \leq \frac{L-2\ell}{2|v(k)|}$ which is the same delay time as found in the swap gate, for which the velocity is 2. This corresponds to the time at which the quasiparticles of velocity $v(k)$, produced at points a distance $\pm L/2$ from the midpoint of $A$, and which are initially traveling towards $A$, both enter the subsystem, see Figure 4. The use of the time variable $\tau_k$ can then be understood by noting that there are also quasiparticles produced, which initially travel away from the subsystem but which enter $A$ after a time which is delayed by $\frac{L}{v(k)}$. Accordingly, we should replace $t \to t \bmod L/|v(k)|$ to obtain an expression valid at arbitrary times. This replacement of the time arises explicitly from the stationary phase calculation by determining the range of validity of the saddle point solutions and how their upper limit depends upon $v(k), t, L$. More details on this are presented in appendix A. In Figure 5, we plot $\langle N_A^2(t)\rangle$ comparing the exact expression, which we evaluate numerically, with the one obtained from (68), finding excellent agreement.

The same analysis can also be carried out when the subsystem consists of both $A$ and its mirror $A^M$ allowing us to obtain the quasiparticle prediction for

$$\begin{aligned}
\sigma_{A\cup A^M}^2(t) &= \langle N_A^2(t)\rangle + \langle N_{A^M}^2(t)\rangle + \langle N_A(t)N_{A^M}(t)\rangle + \langle N_{A^M}(t)N_A(t)\rangle - 4\ell^2 \\
&= 2\langle N_A^2(t)\rangle + 2\langle N_A(t)N_{A^M}(t)\rangle - 4\ell^2 \,.
\end{aligned} \tag{70}$$

Where, in going to the second line, we have used the translational invariance of both the initial state and the Hamiltonian. The second term, $\langle N_A(t) N_{A^M}(t) \rangle$, is new and needs to be considered separately using the multi-dimensional stationary phase approximation. Leaving the details to the appendix, we find that,

$$\sigma^2_{A \cup A^M}(t) = 2\ell - \int_{-\pi}^{\pi} \frac{dk}{2\pi} \Big[ 2 \max\big(0, 2\ell - \big|2\tau'_k v(k) - L\big|\big) + \max\big(0, 2\ell - \big|2\tau'_k v(k)\big|\big)$$
$$+ \max\big(0, 2\ell - \big|2\tau'_k v(k) - 2L\big|\big)\Big], \tag{71}$$

$$\tau'_k \equiv t \bmod \frac{L}{2|v(k)|}. \tag{72}$$

Here, we can note the presence of three types of counting functions, the one which appears in (68), the standard one, and a new one involving a shift of $L$. The first of these arises from the first term in (70). The appearance of the standard one could be anticipated since we consider a subsystem which includes both $A$ and its mirror. Therefore, at short times the disjoint subsystem contains both members of a correlated quasiparticle pair and behaves similarly to the usual quench scenario. The final counting function merely accounts for the fact that upon traversing half the system a quasiparticle pair appears in the same configuration as it was initially but with the positions of each member of the pair exchanged. It should be noted that, the time variable, $\tau'_k$, differs from the case of the single contiguous subsystem with the period being half that of a single interval. Previously, the need for $\tau_k$ arose from the fact that there were quasiparticle pairs which traveled the long way round the system before entering $A$. In this case since $A$ and $A^M$ are diametrically positioned, one can view the system as being effectively halved in size leading to $\tau_k \to \tau'_k$, see Figure 6. As detailed in the appendix, this can be derived explicitly by considering the range of validity of the stationary phase approximation. In Figure 5 we plot the charge fluctuations using (71) and compare it to the exact numerical result obtained through the correlation matrix, finding excellent agreement, even for very small subsystems.

### 3.3 Entanglement dynamics

Having discerned the structure of the quasiparticle dynamics by studying the fluctuations of charge inside the subsystem, we can now obtain the quasiparticle prediction for the entanglement entropy and the mutual information. The remaining piece of information which is required, is to understand the contribution of each quasiparticle pair to the entropy. For this, it is useful to once again recall how this works in the standard scenario. When quenching from a lowly entangled state which produces locally correlated pairs of quasiparticles, the subsystem initially contains both members of a pair and so these cannot generate entanglement between $A$ and $\bar{A}$. After some time however, the subsystem contains one and only one member of a correlated pair whereupon the entanglement entropy receives a contribution equal to the thermodynamic entropy of the corresponding quasiparticle. The entanglement therefore increases as the number of pairs which are shared between the subsystem and its complement increases. This continues until the subsystem only contains quasiparticles whose partner is outside the subsystem, leading to a saturation of $S_A(t)$ to a value which is proportional to the subsystem size.

In our case, the region $A$ is maximally entangled with $\bar{A}$ and initially contains only a single member of a correlated pair. Thus each pair which is shared between $A$ and $\bar{A}$ contributes $\log(2)$ to the entanglement. After some time, the subsystem will not only contain a quasiparticle whose partner is in $\bar{A}$ but also some whose partner is inside $A$ as well. Such pairs, which are

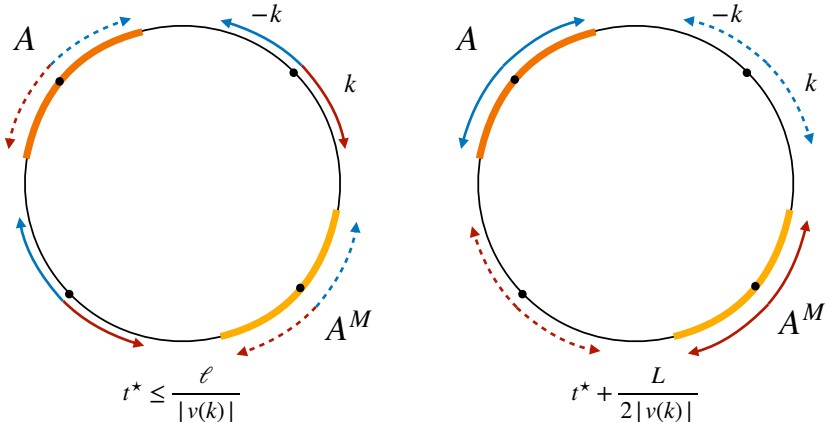

Figure 6: A depiction of the quasiparticle dynamics for the disjoint subsystem $A \cup A_M$. Quasiparticle pairs, depicted by red and blue, dashed and solid arrows, are emitted from each point in space. Correlations exist only between quasiparticles represented by the same color and style, dashed or solid. On the left, we depict the situation for some time $t^\star \leq \frac{\ell}{|v(k)|}$. The disjoint subsystem contains two correlated pairs, denoted by the dashed arrows of different colours. On the right, we depict the situation at the time $t^\star + L/2|v(k)|$. Here we see that the disjoint subsystem still contains two correlated pairs, denoted by the solid arrows. In this instance the correlated pairs are contained entirely within $A$ or $A_M$ but, from the point of view of charge fluctuations or entanglement, the contributions are the same as on the left, necessitating the use of $\tau'_k$ in (71).

completely inside the subsystem, cannot contribute positively to the entanglement and so there is an overall decrease in entanglement of $2 \log(2)$. This value of the drop arises due to the fact we have replaced two entangling quasiparticles, each contributing $\log(2)$, with a single pair giving no contribution. From this picture, we then find that

$$S_A(t) = 2\ell \log(2) - 2\log(2) \int_{-\pi}^{\pi} \frac{dk}{2\pi} \max\left(0, 2\ell - |2\tau_k v(k) - L|\right). \tag{73}$$

This result can also be derived directly using the multi-dimensional stationary phase approximation. In Figure 7 we plot this expression, comparing against the numerically exact result obtained from (60) with excellent agreement, even for small subsystem sizes. We see that as expected $S_A(t)$ remains constant up until $t = \frac{L-2\ell}{4}$ after which time it begins to decrease linearly in $t$. Around $t = L/4$ the entanglement entropy starts to rise again before experiencing another drop around $t = \frac{3L-2\ell}{4}$ followed by further drops and peaks. The first of these features, the time delay prior to the linear decrease, has been anticipated in the previous section and corresponds to the first time a correlated pair of quasiparticles enters the subsystem. The second feature, wherein the $S_A(t)$ begins to rise again can be understood by noting that after $t = L/2|v(k)|$ correlated pairs have traversed a quarter of the system and now reside at the same point in space. From this time on, from the point of view of this pair, the dynamics resembles the typical scenario where correlated pairs evolve from the same point. Accordingly, this leads to an increase in entanglement entropy. The final feature, the drop beginning at $t = \frac{3L-2\ell}{4}$, arises in the expression (73) from the replacement $t \to \tau_k$ and can be understood as being the contribution to the dynamics of the quasiparticle pairs which took the long way

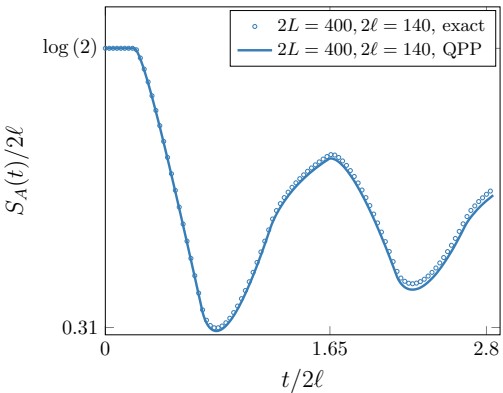 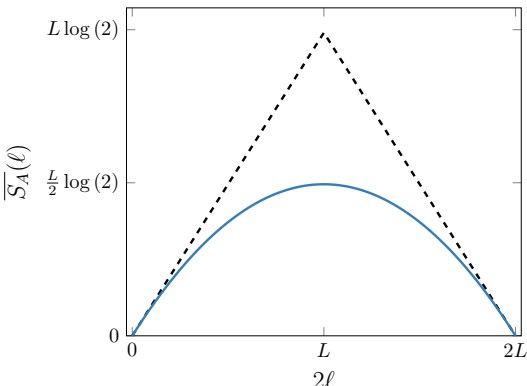

Figure 7: Left:$S_A(t)$ for total system size $2L = 400$ and subsystem size $2\ell = 140$, with the blue curve being the exact calculation and the orange curve using the quasiparticle picture. Right: The time averaged entanglement entropy $\overline{S_A}(\ell)$ as a function of subsystem size (solid blue). Also shown is the initial value $S_A(0)$ (dashed black) which coincides with the time averaged entanglement under random unitary of generic dual unitary dynamics in the large $q$ limit.

around the system before entering $A$. All subsequent revivals and depletions can be understood through these mechanisms while the fact that the increases and decreases are not strictly linear comes about due to the non-linear dispersion of the model.

We may explore the average entanglement entropy of the system by computing the time averaged entanglement at large time as a function of subsystem size,

$$\overline{S_A}(\ell) = \lim_{T \to \infty} \int_0^T \frac{\mathrm{d}t}{T} S_A(t). \tag{74}$$

This quantity is tricky to compute analytically, however, it is straightforward to understand within the quasiparticle picture. In particular, at a generic point in time, the probability that a given quasiparticle within $A$ has a partner which is not within the subsystem is given by $1 - \ell/L$. The entanglement contribution of such a pair is $\log(2)$ and there are on average $\ell$ quasiparticles inside $A$. Thus, after averaging over time, one arrives at,

$$\overline{S_A}(\ell) = \log(2)\frac{\ell(L - \ell)}{L}, \tag{75}$$

where we have used $S_A(t) = S_{\bar{A}}(t)$ to extend the result beyond $\ell > L/2$. Numerically integrating (74) for large, but finite $T$, one finds exact agreement with the above result. We plot this in Figure 7, including also $S_A(0)$ for comparison, which is also the time averaged entanglement for a system undergoing random unitary or generic dual unitary dynamics. This result is reminiscent of the Page curve for Gaussian random states [93]. In fact (75) can be seen as a special case of this wherein one averages over random states which have a quasiparticle pair structure.

The same strategy can be used to obtain the mutual information. The key new ingredient here is the behaviour of $S_{A \cup A^M}(t)$. By using the insight gained from the charge fluctuations as

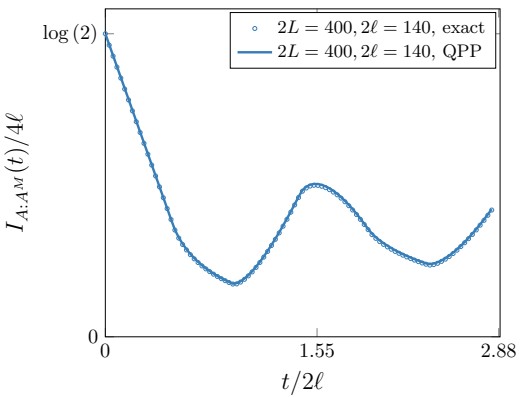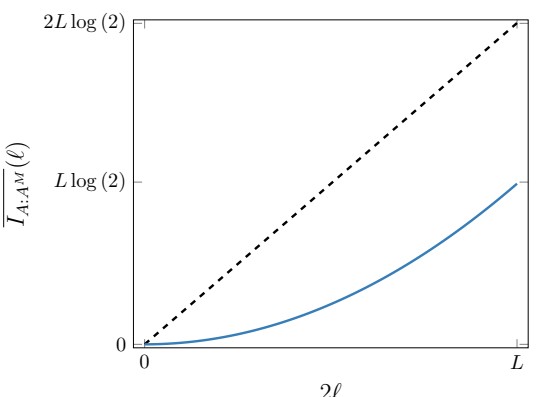

Figure 8: Left: $I_{A:A_M}(t)$, for total system size $2L = 400$ and subsystem size $2\ell = 140$, with the symbols being the exact numerical result and the blue curve using the quasi-particle picture. Right: The time averaged mutual information $\overline{I_{A:A_M}}(\ell)$ as a function of $\ell \leq L/2$ (solid blue). For comparison, we also plot the initial value $I_{A:A^M}(0)$ (dashed black).

well as the single interval case discussed above we determine that

$$S_{A \cup A^M}(t) = 2\log(2) \int_{-\pi}^{\pi} \frac{dk}{2\pi} \left[ \min\left(2\ell, \left|2\tau'_k v(k)\right|\right) - 2\max\left(0, 2\ell - \left|2\tau'_k v(k) - L\right|\right) \right. \tag{76}$$
$$\left. -2\max\left(0, 2\ell - \left|2\tau'_k v(k) - 2L\right|\right) \right].$$

Note that here, the entanglement starts at zero and then increases linearly in time due to the first term. After $t = \frac{L - 2\ell}{4}$ and $\frac{L - 2\ell}{2}$ however the second and third terms respectively become nonzero and contribute to an overall decrease in the entanglement.

Combining (76) and (73), we obtain the mutual information. In Figure 8 we plot the analytic result and compare to the exact numerical data with excellent agreement. From this we see that, as in the circuit dynamics, $I_{A:A^M}(t)$ experiences an initial linear decrease in entanglement. For the chosen subsystem sizes however, this never reaches zero and $I_{A:A^M}(t)$ starts to experience revivals. This is in contrast to the chaotic circuits of the previous section in which the mutual information vanishes for $t > \ell/2$. Using (75) one can also obtain the time averaged behvaiour of $I_{A:A^M}(t)$. Concentrating on the case where $A$ and $A^M$ are not overlapping, $\ell \leq L/2$, we find

$$\overline{I_{A:A^M}}(\ell) = 2\overline{S_A}(\ell) - \overline{S_{A \cup A^M}}(\ell) \tag{77}$$
$$= 2\overline{S_A}(\ell) - \overline{S_A}(2\ell),$$

which is also plotted in Figure 8 along with $I_{A:A^M}(0)$ for comparison.

## 3.4 Interacting model

When $U \neq 0$ the model is interacting and we can no longer rely upon exact free fermion techniques to calculate $S_A(t)$ or $I_{A:A^M}(t)$. Being integrable, however, the model still admits a description in terms of quasiparticles. Therefore, we can adapt the quasiparticle picture we have derived in the previous section to the interacting case. The quasiparticle content of the model is naturally more complicated than in the free case and we provide a complete review in appendix B. The important points are, however, the same as in the free case. Namely, the

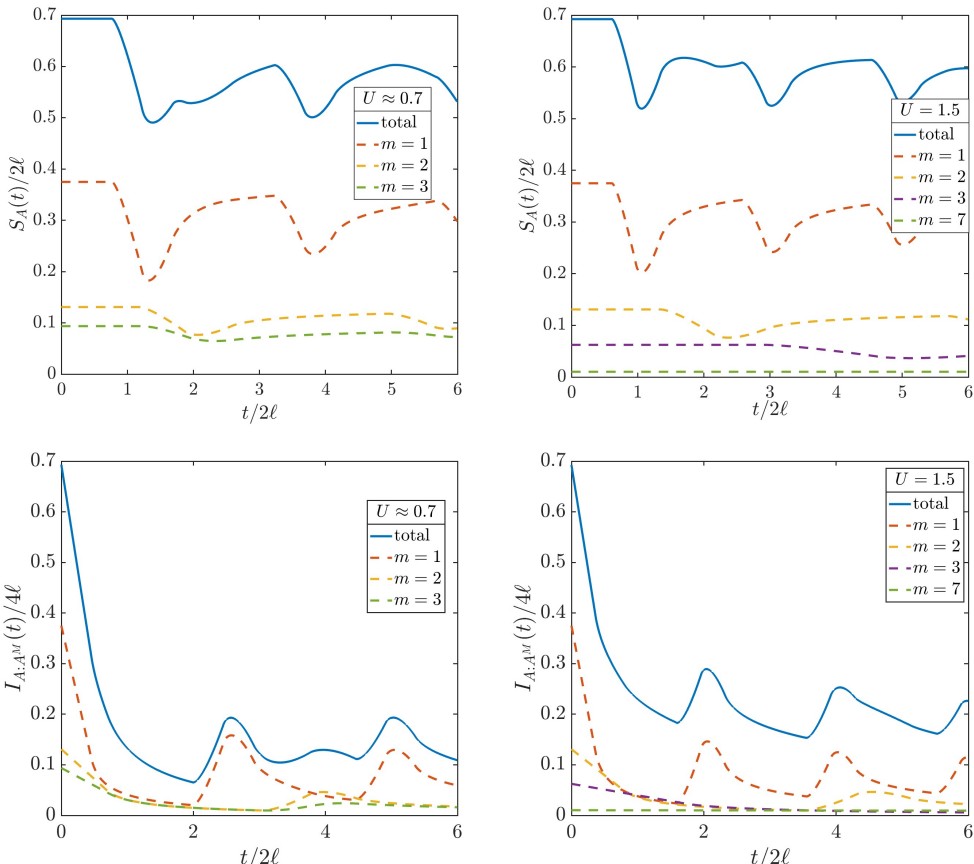

Figure 9: Left: $S_A(t)/2\ell$ (Top) and $I_{A:A^M}(t)/4\ell$ (bottom) for $2L = 160, 2\ell = 30$ obtained from the quasiparticle picture in the gapless regime, using $U = \cos(\pi/4) \approx 0.7$ (solid lines). We also plot the contributions of the quasiparticle species with $m = 1, 2, 3$ (dashed lines). Right: $S_A(t)/2\ell$ (Top) and $I_{A:A^M}(t)/4\ell$ (bottom) for $2L = 160, 2\ell = 30$ in the gapped regime with $U = 1.5$ (solid lines). We also plot the contributions of the quasiparticle species with $m = 1, 2, 3, 7$ (dashed lines).

quench produces pairs of quasiparticles which propagate ballistically throughout the system and to find the entanglement entropy we should simply count the contribution of each quasiparticle pair.

In the interacting case there are many different types of quasiparticle. These can be thought of as bound states of fermionic excitations and are labeled by a discrete index $m = 1, \ldots, M$, the species index, and a continuous variable $\lambda \in [-\Lambda, \Lambda]$ known as the rapidity [86]. The latter can be understood as being the counterpart to the momentum, $k$, in the free model, and the former is, roughly, the number of fermions which are bound together. The specific values of $M$ and $\Lambda$ depend upon the parameter regime of the model, which broadly spits into two distinct regimes: the gapless regime for $|U| \leq 1$ and the gapped regime $|U| > 1$. In the former, at special points in parameter space where $U = \cos\left(\frac{\pi}{p+1}\right)$ for integers $p \geq 1$, there are a finite number of quasiparticle species, $M = p + 1$, meaning that a maximum number of fermions can combine to form a single bound state. In the latter regime, however, $M = \infty$ and bound states of arbitrary size can form. In either case, the result is that

$$S_A(t) = 2\ell \log(2) - 2 \sum_{m=1}^{M} s_m \int_{-\Lambda}^{\Lambda} d\lambda \, \rho_n^t(\lambda) \max\left(0, 2\ell - \left|2\tau_{\lambda,m} v_m(\lambda) - L/2\right|\right), \tag{78}$$

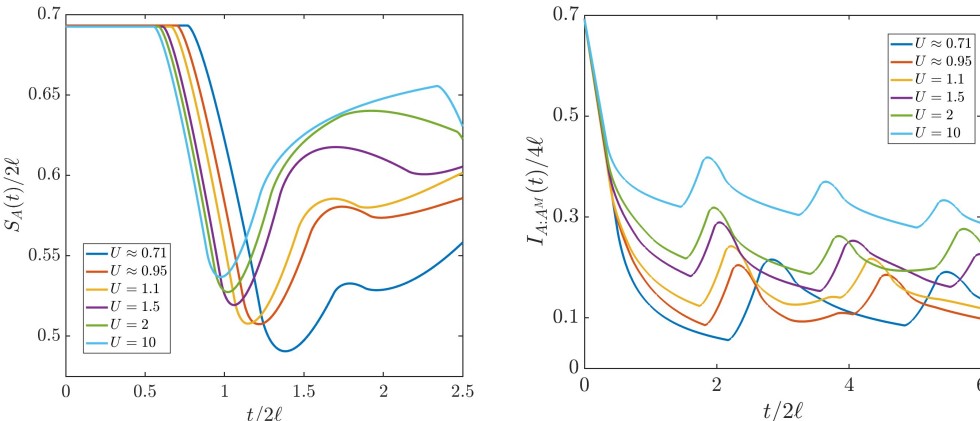

Figure 10: Left: The behaviour of $S_A(t)/2\ell$ as a function of time for different values of $U$ in both the gapless and gapped cases. As the interactions strength is increased the initial decay occurs earlier and earlier. Right: The behaviour of $I_{A:A^M}(t)/4\ell$ as a function of time for different values of $U$.

where $s_m$ is the quasiparticle contribution to entanglement, $v_m(\lambda)$ is the quasiparticle velocity, $\rho_m^t(\lambda)$ is the density of states of the quasiparticles and $\tau_{\lambda,m}$ the corresponding periodic time. Each of these quantities can be calcuated exactly using thermodynamic Bethe ansatz techniques [86] and the fact that the crosscap state is integrable [54]. A key simplifying component in this analysis is that the initial state locally resembles the infinite temperature state, whose properties are well known and allow one to determine $s_m$ analytically. We provide the details of these calculations as well as those of $v_m(\lambda)$ and $\rho_m^t(\lambda)$ in appendix B. A similar generalization also leads to an expression for the mutual information, $I_{A:A^M}(t)$.

In Figure 9 we plot $S_A(t)$ and $I_{A:A^M}(t)$ for different interactions strengths using (78). On the top left of the plot we show $S_A(t)$ (solid line) for $U = \cos(\pi/4) \approx 0.7$ as well as the individual contributions of the $m = 1, 2, 3$ quasiparticles species (dashed lines). Here we see that each quasiparticle species has a contribution which is qualitatively the same as that seen in the free case seen in Figure 7. They exhibit a constant initial period, followed by a sudden decay and later growth which is then repeated. The time scales differ between the species however, which arises due to the fact that they have different velocities. Indeed, the more fermions which make up the bound state the slower it is, i.e. $v_{m_1}(\lambda) < v_{m_2}(\lambda)$ for $m_1 > m_2$ while at the same time the smaller its contribution to the entanglement. The summation over the different species then results in a series of new peaks and troughs which do not appear in the free case. For example, the small peak around $t/2\ell \approx 2$ results from the growth of the $m = 1$ contribution and the simultaneous decay of the $m = 2, 3$ parts. A similar behaviour is also seen in $I_{A:A^M}(t)$ which is plotted on the bottom left. On the right, we show $S_A(t)$ in the gapped regime using $U = 1.5$. Here, there are bound states of all sizes and we display only the contributions of a few (dashed lines) as well as the total value (solid line). We see, also here, that the individual quasiparticles have the same qualitative features as in the gapless regime. One can note, however, that the time scale for the $m = 1$ species to start evolving has decreased whereas, those for $m > 1$ have become longer. This can be attributed to the slowing down of bound states in this regime. In fact, in the limit $U \to \infty$ one finds that $\max[v_1(\lambda)] \to 3$ while $\max[v_{m>1}(\lambda)] \to 0$ and the bound states become effectively non-dynamical on ballistic scales [94].

In Figure 10 we plot $S_A(t)$ and $I_{A:A^M}(t)$ for different values of the interactions strength in both the gapped and gapless regimes. We can note that the behaviour is continuous as one crosses between regimes. As with the free case one can also examine the long time average of these quantities. The same phenomenology applies and one arrives at the result (75), however in the gapped regime this is difficult to verify numerically due to the presence of the large, slow bound states which require one to average over very large $T$. It is possible to verify, however, that as $T$ increases one approaches the result (75).

## 4 Conclusions

In this paper, using a mix of analytic calculations and numerics, we have studied the dynamics of the entanglement entropy and mutual information in quenches from a class of long-range-correlated states known as crosscap states. In particular, the initial states consisted of generalized EPR pairs that sit on antipodal points of a periodic qudit chain, and the time evolution was generated by either brickwork quantum circuit or Hamiltonian dynamics, which included both integrable and chaotic examples. For the circuit dynamics, we considered both random unitary and dual unitary evolution, while for the Hamiltonian system, we studied an interacting fermion chain equivalent to the XXZ model. Depending on the nature of the dynamics, we witnessed two distinct patterns of behaviour. For integrable systems, we saw that initially the entanglement entropy remained constant but began to decrease after a time which was dependent on the subsystem and total system sizes. This was followed by an increase and then a subsequent decrease, with this pattern repeating periodically. In contrast, for chaotic systems the entanglement entropy stayed constant. At early times, however, for both integrable and chaotic dynamics, the mutual information experienced a linear decrease in time. In chaotic systems, this continued until the mutual information vanished and remained that way for all times. However, in the integrable case, the mutual information exhibited a series of revivals. In integrable models, both the entanglement entropy and mutual information possess non-trivial time averaged profiles as a function of subsystem size. After suitable modifications, we showed that these behaviours could be described using the established effective theories of entanglement dynamics: the quasiparticle and membrane pictures.

A natural generalization of this work would be to consider initial states in which more than two sites, sitting at distant points on the chain, are entangled. For integrable models, one could expect in that case the creation of correlated multiples of particles rather than just pairs, leading to more complex dynamics. It would also be interesting to examine properties of such systems other than their entanglement, e.g. the restoration of symmetry through the entanglement asymmetry [26–29]. Alternatively, in the free case one can obtain a quasiparticle description of the reduced density matrix itself, following a recent approach to quenches from lowly entangled states [31, 32]. Measures of mixed-state entanglement, such as the negativity, could also be explored by extending the results derived for lowly entangled initial states [18, 19, 95]. Finally, for lowly entangled initial states, one might expect features of the quasiparticle picture to persist on certain timescales in the presence of weak integrability breaking terms. For quenches from crosscap states however, the entanglement for a single interval only shows dynamics on scales proportional to the system size. Therefore, it is possible that weak integrability breaking could see a significant departure from the quasiparticle picture and features similar to those described by the membrane picture.

## Acknowledgments

We thank Filiberto Ares and Andrei Rotaru for useful discussions on the topic.

**Funding information**   PC and CR acknowledge support from ERC under Consolidator Grant number 771536 (NEMO) and from European Union - NextGenerationEU, in the framework of the PRIN Project HIGHEST number 2022SJCKAH_002.

## A   Stationary phase approximation derivation for the single and the disjoint intervals

**Single interval:**   In this appendix, we present in detail the derivation of the quasiparticle picture presented in section 3 using the stationary phase approximation [87]. By applying this method to the exact expressions, one is lead to an analytic result that is then interpreted through the quasiparticle picture. The first step in this procedure, is to consider $\langle N_A^2(t) \rangle$ from equations (64) and (67) and convert the sums that go over the subsystem sites, into integrals, using the following identity

$$e^{-i(2\ell+1)\frac{k}{2}} \sum_{x=1}^{2\ell} e^{ixk} = \ell \int_{-1}^{1} d\xi \frac{k/2}{\sin(k/2)} e^{i2\ell\xi\frac{k}{2}} \,. \tag{A.1}$$

Here we see that the sum over the subsystem sites indexed by the discrete variable $x$, has been exchanged with an integral over a continuous variable $\xi \in [-1, 1]$. The difference from the main text, is that we present the problem solved from a generic initial time $t_0$ until a time $t$, hence instead of $t$ we will have $t - t_0$. By using equation (64), we solve (67) for $\langle N_A^2(t) \rangle$ and apply the identity for the two indices $x, y \in [1, 2\ell]$, obtaining

$$\begin{aligned}
\langle N_A^2(t) \rangle = &\ell^2 - \ell/2 - \ell^2 \int_{[-\pi,\pi]^2} \frac{dk_1 dk_2}{(2\pi)^2} e^{2i(t-t_0)(\epsilon(k_2)-\epsilon(k_1))} \\
&\times \int_{[-1,1]^2} d\xi_1 d\xi_2 \frac{(k_1-k_2)/2}{\sin((k_1-k_2)/2)} e^{i\ell(k_2-k_1)(\xi_1-\xi_2)+iL(k_1-k_2)} \,.
\end{aligned} \tag{A.2}$$

The integrand depends on the momenta and on the difference $\xi_1 - \xi_2$, meaning that it can be simplified by a change of coordinates to $(\zeta_0, \zeta_1) = (\xi_1, \xi_2 - \xi_1)$, resulting in

$$\begin{aligned}
\langle N_A^2(t) \rangle = &\ell^2 - \ell/2 - \ell^2 \int_{[-\pi,\pi]^2} \frac{dk_1 dk_2}{(2\pi)^2} e^{2i(t-t_0)(\epsilon(k_2)-\epsilon(k_1))} \\
&\times \int_{\mathcal{D}} d\zeta_0 d\zeta_1 \left( \frac{(k_1-k_2)/2}{\sin((k_1-k_2)/2)} \right)^2 e^{-i\ell(k_1-k_2)\zeta_1 + iL(k_1-k_2)} \,.
\end{aligned} \tag{A.3}$$

Our interest lies in the scaling limit where $L, \ell, t \to \infty$ keeping $\zeta = \frac{t}{\ell}$ fixed. In this limit the integral is estimated using the stationary phase approximation. The point in which the phase is stationary satisfies the condition $k_1 \approx k_2$ since the phase of the exponential depends only on the difference $k_2 - k_1$, a step known as the "localization rule" [87], which also means that $\lim_{k_1 \to k_2} \left( \frac{(k_1-k_2)/2}{\sin((k_1-k_2)/2)} \right)^2 = 1$. Since the integrand depends only on the $\zeta_1$ coordinate, we are able to integrate over $\zeta_0$, obtaining the integration measure $\mu(\zeta_1)$, which along with the above

simplification lead us to

$$
\begin{aligned}
\langle N_A^2(t)\rangle = &\ell^2 - \ell/2 - \ell^2 \int_{[-\pi,\pi]^2} \frac{dk_1 dk_2}{(2\pi)^2} e^{2i(t-t_0)(\epsilon(k_2)-\epsilon(k_1))} \\
&\times \int_{\mathcal{D}} d\zeta_1 \mu(\zeta_1) e^{-i\ell(k_1-k_2)\zeta_1 + iL(k_1-k_2)},
\end{aligned}
\tag{A.4}
$$

where this measure function is known and takes the form

$$
\mu(\zeta_1) = \begin{cases} 2 - |\zeta_1|, & \text{if } |\zeta_1| < 2, \\ 0, & \text{if } |\zeta_1| > 2. \end{cases}
\tag{A.5}
$$

The double integral over $k_2$ and $\zeta_1$, is of the type $\int_{\mathcal{F}} d^N \vec{x} p(\vec{x}) e^{i\ell q(\vec{x})}$, which, in the saddle point approximation for large $\ell$ is given by

$$
\int_{\mathcal{F}} d^2 \vec{x} p(\vec{x}) e^{i\ell q(\vec{x})} \approx \left(\frac{2\pi}{\ell}\right) p(\vec{x}_0) \frac{1}{\sqrt{\det(A)}} e^{iq(\vec{x}_0) + \frac{i\pi\sigma_A}{4}},
\tag{A.6}
$$

with $A$ being the Hessian matrix $A_{m,n} = \frac{\partial}{\partial_{x_n}} \frac{\partial}{\partial_{x_m}}$, $\det(A)$ its determinant and $\sigma_A$ the number of positive minus the number of negative eigenvalues. The point $\vec{x}_0$ is the saddle point of the integral, that satisfies the saddle point condition $\nabla q(\vec{x})|_{\vec{x}_0} = 0$,

with enumeration of coordinates being $m, n = \zeta_1, k_2$. For our case the functions $p(\vec{x})$ and $q(\vec{x})$ are given by

$$
p(\vec{x}) = \mu(\zeta_1); \qquad q(\vec{x}) = (k_1 - k_2)\zeta_1 + \frac{L}{\ell}(k_1 - k_2) + 2\frac{(t-t_0)}{\ell}(\epsilon(k_2) - \epsilon(k_1)),
\tag{A.7}
$$

which leads us to the saddle point conditions being satisfied at the saddle point $(\bar{\zeta}_1, \bar{k}_2)$ that can be determined to be

$$
\bar{k}_2 = k_1, \qquad \bar{\zeta}_1 = 2\frac{(t-t_0)}{\ell} \frac{d}{dk_2} \epsilon(k_2)|_{k_2 = \bar{k}_2} - \frac{L}{\ell}.
\tag{A.8}
$$

The Hessian matrix $A$ has $\sigma_A = 0$ and $\det(A) = -\frac{1}{4}$, giving us

$$
\langle N_A^2(t)\rangle \approx \ell^2 + \ell/2 - \ell \int_{-\pi}^{\pi} \frac{dk}{2\pi} \max\left(0, 1 - \left|2\frac{(t-t_0)}{\ell} v(k) - \frac{L}{\ell}\right|\right),
\tag{A.9}
$$

which is a result that has a quasiparticle picture interpretation, with a corresponding counting function that counts pairs, that we call $R(k, t-t_0) = \max\left(0, 1 - \left|\frac{(t-t_0)}{\ell} v(k) - \frac{L}{\ell}\right|\right)$.

**The quasiparticle structure from the reconfiguration times of the modes:** Recurrences of the quasiparticle structure are uncovered by examining the solution of the saddle point equations. Studying the problem for an initial time $t_0$ up to a time $t$, we can write the saddle point equation of $\bar{\zeta}_1$ of a particular mode $k$ as follows

$$
t - t_0 = \frac{\bar{\zeta}_1 \ell + L}{2L} \frac{L}{v(k)}.
\tag{A.10}
$$

Since we have the definition $\zeta_1 = \xi_1 - \xi_2$ (where $\xi_1, \xi_2 \in [-1, 1]$), then $|\zeta_1| \leq 2$ and we moreover know that $\ell$ is physically restricted to $\ell \leq \frac{L}{2}$, there is an upper bound for this time window $(t - t_0)(k)$ equal to

$$
\delta t_{max}(k) = (t - t_0)_{max}(k_1) = \frac{L}{|v(k)|},
\tag{A.11}
$$

which is finite for the modes $k$ with nonzero velocity, $v(k) \neq 0$ due to $L$ being finite. Thus, the saddle point solution we presented in equation (A.8) for each mode $k$ stops being valid after this window is surpassed and the equations need to be solved again for a new initial time. This leads to the repetition of the counting function evolution for the $k$ modes in that time window, starting again from its initial value, so we make the replacement

$$t - t_0 \to \tau_k = (t - t_0) \bmod \frac{L}{|v(k)|}, \tag{A.12}$$

leading to the final QPP expression,

$$\langle N_A^2(t) \rangle = \ell^2 + \ell/2 - \ell - \ell \int_{-\pi}^{\pi} \frac{\mathrm{d}k}{2\pi} \max\left(0, 1 - \left|2\frac{\tau_k}{\ell}v(k) - \frac{L}{2\ell}\right|\right). \tag{A.13}$$

**Disjoint intervals:** The same procedure is applied to derive the quasiparticle picture for the disjoint interval case, but for each one of the three terms of $\langle N_{A \cup A^M}^2(t) \rangle$, which we remind here

$$\langle N_{A \cup A^M}^2(t) \rangle = \langle N_A^2(t) \rangle + \langle N_{A^M}^2(t) \rangle + \langle N_A(t)N_{A^M}(t) \rangle + \langle N_{A^M}(t)N_A(t) \rangle. \tag{A.14}$$

The first two terms $\langle N_A^2(t) \rangle = \langle N_{A^M}^2(t) \rangle$, give the same quasiparticle counting function as analyzed in section A, which refer to the case of the single interval, given by equation (A.13). The difference comes from the other two terms, which turn out to give two new counting functions. Each of these terms codifies a different time behavior of the subsystem that is comprised by the union of the two disjoint intervals $A$ and $A^M$. Specifically we have the term $\langle N_A(t)N_{A^M}(t) \rangle$ that reads

$$\langle N_A(t)N_{A^M}(t) \rangle = \ell^2 + \ell/2 - \ell \int_{-\pi}^{\pi} \frac{\mathrm{d}k}{2\pi} \max\left(0, 1 - \left|2\frac{\tau}{\ell}v(k) - L\right|\right), \tag{A.15}$$

and the term exchanging the order of $N_A(t)$ and $N_{A^M}(t)$

$$\langle N_{A^M}(t)N_A(t) \rangle = \ell^2 + \ell/2 - \ell \int_{-\pi}^{\pi} \frac{\mathrm{d}k}{2\pi} \max\left(0, 1 - \left|2\frac{\tau}{\ell}v(k)\right|\right). \tag{A.16}$$

Note that microscopically the above terms are equal, however, we should take care when passing to the thermodynamic limit. Hence, we find three in general different recurrence times per mode $k$, which are

$$\tau_{L/2} = \frac{L}{|v(k)|}, \qquad \tau_L = \frac{3L}{2|v(k)|}, \qquad \tau_0 = \frac{L}{2|v(k)|}, \tag{A.17}$$

and refer to the validity of the three different saddle point solution of the aforementioned terms for the mode $k_1$. But since the time of the quench is the same for all three, the smallest one is when the evolution has to be updated, which is always $\tau_0$ and when we substitute it indeed we get really good agreement with numerics and can be also intuitively understood as is explained in the caption of Figure 6.

## B Thermodynamic Bethe ansatz description of $H_U$

The Hamiltonian $H_U$ is integrable and simply related to the XXZ spin chain by a Jordan-Wigner transformation. The eigenstates and spectrum of $H_U$ are known exactly [84, 85] allowing one

to determine some of its properties exactly. It has two regimes, a gapless regime for $|U| < 1$ and a gapped regime for $|U| > 1$. At $U = 1$ it has an enhanced $SU(2)$ symmetry, although we do not investigate directly this point. In the thermodynamic limit the system admits a description in terms of the thermodynamic Bethe ansatz which we now briefly recap [86]. The system supports a set of stable quasiparticle excitations specified by a species index $m = 1, \ldots, M$ and a rapidity variable $\lambda \in [-\Lambda, \Lambda]$. The number of species, $M$, and bounds on the rapidity, $\Lambda$ depend on the regime of the model and the particular value of the interaction parameter for instance, at $U > 1$ $M = \infty$ and $\Lambda = \pi/2$.

A stationary state of the system is described through a set of distributions for these quasiparticles, in particular $\vartheta_m(\lambda) \in [0, 1]$ is the occupation function of quasiparticle of species $m$ at rapidity $\lambda$, $\rho_m^t(\lambda)$ is their density of states and $\rho_m(\lambda) = \vartheta_m(\lambda)\rho_m^t(\lambda)$ is the distribution of occupied modes. These distributions are coupled to each other via a set of integral equations whose form also depends on the parameter regime. Each quasiparticle has a bare velocity which is dressed by the presence of interactions, we denote this by $v_m(\lambda)$. We now present details in each of the regimes, employing the following notation

$$(f \star g)(\lambda) = \int_{-\Lambda}^{\Lambda} d\mu \, f(\lambda - \mu) g(\mu). \tag{B.1}$$

**Gapless regime:** In the gapless regime we parameterize the interaction via the parameter $\gamma$ defined by $U = \cos(\gamma)$ and moreover we restrict to the simplest possible case wherein $\gamma = \frac{\pi}{p+1}$. With this choice there are $M = p + 1$ quasiparticle species also known as strings, the first $p$ of which have bare charge or string length $q_m = m$ while the last has $q_{p+1} = 1$. These strings can be interpreted as bound states of $q_m$ fermions. For a given stationary state defined by a set of occupation functions $\vartheta_m(\lambda)$ the density of states $\rho_m^t(\lambda)$ and density of occupied modes $\rho_m(\lambda)$ are related by the integral equations (we drop the explicit $\lambda$ dependence to lighten the notation),

$$\rho_1^t = s + s \star \left[ (1 - \vartheta_2)\rho_2^t + \delta_{p,2}\rho_{p+1} \right], \tag{B.2}$$

$$\rho_m^t = s \star \left[ (1 - \vartheta_{m-1})\rho_{m-1}^t + (1 - \vartheta_{m+1})\rho_{m+1}^t + \delta_{p,m+1}\rho_{p+1} \right], \quad 1 < m < p, \tag{B.3}$$

$$\rho_p^t = \rho_{p+1}^t = s \star \left[ (1 - \vartheta_{p-1})\rho_{p-1}^t \right]. \tag{B.4}$$

with $s(\lambda) = \mathrm{sech}(\pi\lambda/\gamma)/(2\gamma)$, and the bound on rapidities now being $\Lambda = +\infty$. A similar set of integral equations determine the quasiparticle velocity

$$v_1 \rho_1^t = -\frac{\sin\gamma}{2} \partial_\lambda s + s \star \left[ (1 - \vartheta_2)v_2 \rho_2^t + \delta_{p,2}v_{p+1}\rho_{p+1} \right], \tag{B.5}$$

$$v_m \rho_m^t = s \star \left[ (1 - \vartheta_{m-1})v_{m-1}\rho_{m-1}^t + (1 - \vartheta_{m+1})v_{m+1}\rho_{m+1}^t + \delta_{p,m+1}v_{p+1}\rho_{p+1} \right], \quad 1 < m < p, \tag{B.6}$$

$$v_p \rho_p^t = \rho_{p+1}^t v_{p+1} = s \star \left[ (1 - \vartheta_{p-1})v_{p-1}\rho_{p-1}^t \right]. \tag{B.7}$$

The occupation functions are required as inputs for all these equations. These are found by noting that the reduced density matrix of the subsystem is the infinite temperature state. The occupation functions for this state are [86]

$$\vartheta_m = \frac{1}{(m+1)^2}, \quad 1 \le m < p, \quad \vartheta_p = 1 - \vartheta_{p+1} = \frac{1}{p+1}. \tag{B.8}$$

The above equations need to be integrated numerically in order to obtain the results on the main text. To do this one must impose a cutoff on $\Lambda$, we choose $\Lambda = 10$ and have checked that results to not vary by increasing this value.

**Gapped regime:** In the gapped regime we instead parametrize the anisotropy by $\eta$ which is defined by $U = \cosh(\eta)$ and unlike in the gapless regime we place no restrictions upon $\eta$. The nature of the strings in this regime changes and as previewed above we have that $M = \infty$ and $\Lambda = \frac{\pi}{2}$. The various distributions governing the stationary states of the model obey the integral equations

$$\rho_1^t = s + s \star \left[ (1 - \vartheta_2)\rho_2^t + \delta_{p,2}\rho_{p+1} \right], \tag{B.9}$$

$$\rho_m^t = s \star \left[ (1 - \vartheta_{m-1})\rho_{m-1}^t + (1 - \vartheta_{m+1})\rho_{m+1}^t \right], \tag{B.10}$$

where now $s(\lambda) = \frac{1}{2\pi} \sum_{k \in \mathbb{Z}} e^{-2ik\lambda} \operatorname{sech}(k\eta)$. Likewise, we have

$$v_1 \rho_1^t = -\frac{\sinh \eta}{2} \partial_\lambda s + s \star \left[ (1 - \vartheta_2)v_2 \rho_2^t \right], \tag{B.11}$$

$$v_m \rho_m^t = s \star \left[ (1 - \vartheta_{m-1})v_{m-1}\rho_{m-1}^t + (1 - \vartheta_{m+1})v_{m+1}\rho_{m+1}^t \right]. \tag{B.12}$$

As in the gapless case the occupation functions can be determined exactly using the integrability of the initial state

$$\vartheta_m = \frac{1}{(m+1)^2}. \tag{B.13}$$

As before these equations need to be integrated numerically to obtain the results of the main text. In this case, we are required to put a cutoff on the number of quasiparticles $M$. Here we choose $M = 40$ and have checked that the results do not change substantially upon increasing this.

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
