# Peer review of "Quench dynamics of entanglement from crosscap states"

_SciPost Physics, doi:SciPost Phys. 19, 132 (2025)_

## Round 1 · Referee Report · Anonymous (Referee 1) · 2025-4-24

Strengths

1- studies an interesting problem in various ways and settings 2- provides analytical results for unusual quenches, both chaotic and integrable 3- is nicely written

Weaknesses

1- Methods are standard 2- Some relevant calculations are left for further work

Report

I believe the journal expectation "Open a new pathway in an existing or a new research direction, with clear potential for multi-pronged follow-up work" is easily met, as the work opens new ways to study quantum quenches from entangled initial states, both for integrable and chaotic models.

The paper provides results in three quite different settings for the same nontrivial initial states and yields insights into differences in integrable and chaotic behaviour. It also discusses how the membrane picture needs to be modified for correlated initial states.

Requested changes

I would like to see the following changes, but I don't insist that all need to be addressed in this paper: - The discussion about how the membrane picture needs to be modified for correlated initial states is a bit vague and imprecise. Extending it, generalizing, and providing more details would help the paper, in my opinion. - I believe the paper would be quite a bit stronger if the recursion equation 2.35 were solved for finite q. I don't see a reason why authors would need to delegate this to further work (if it is doable). Very similar recurrences have been solved in quite a few recent works, so I expect the authors can modify the procedure, for example, https://arxiv.org/pdf/2004.13697. If this does not work, authors can at least comment about why it is more difficult. - Authors can provide a short explanation of why and when Eq. 2.44 holds (apart from citing 54). This approximation gives zero correlations, so in some sense loses all microscopic. -Label under Figure 7. There is probably a typo of->or. They mention random unitaries, but do they mean their q->infinity limit?

Recommendation

Publish (easily meets expectations and criteria for this Journal; among top 50%)

  • validity: top
  • significance: good
  • originality: good
  • clarity: high
  • formatting: excellent
  • grammar: perfect

Author:  Konstantinos Chalas  on 2025-10-14  [id 5933]

(in reply to Report 1 on 2025-04-24)
Category:
answer to question

We would like to thank the referee for carefully considering and reading our manuscript and asking important clarifying questions and moreover giving very insightful comments on how to improve the manuscript.
The replies to the different questions/comments made by the referee are found below and answered in a point-by-point manner.

Comment 1:
The discussion about how the membrane picture needs to be modified for correlated initial states is a bit vague and imprecise. Extending it, generalizing, and providing more details would help the paper, in my opinion.

Our reply:
For the single interval case the entanglement membrane picture is actually the same as originally proposed in references [33-35]. Already in that case it was anticipated that one should incorporate the effects of a highly entangled initial state. Typically, however only product initial states are considered. For the multiple interval case once again only the initial state contribution needs to be modified. This is done by accounting for the non-local nature of the entanglement. We have modified the discussion to clarify the modifications of the membrane picture.

Comment 2:
I believe the paper would be quite a bit stronger if the recursion equation 2.35 were solved for finite q. I don't see a reason why authors would need to delegate this to further work (if it is doable). Very similar recurrences have been solved in quite a few recent works, so I expect the authors can modify the procedure, for example, arxiv 2004.13697. If this does not work, authors can at least comment about why it is more difficult.

Our reply:
Unfortunately the techniques introduced in arXiv:2004.13697 cannot be straightforwardly applied in our case. The main reason for this is that if one includes terms which are subleading in $1/q$, the shape of the resulting diagram can change leading to complications. In the process of solving the recursion relation fully, the appearance of diagrams that are different in shapes for each choice of parameters (subsystem length and time) do not allow to simply apply the aforementioned techniques. Hence we restrict our analysis to leading order in $q$ in the large $q$ limit. We have added a comment on this to the draft.

Comment 3:
Authors can provide a short explanation of why and when Eq. 2.44 holds (apart from citing 54). This approximation gives zero correlations, so in some sense loses all microscopic.

Our reply:
The relation 2.44 arises from standard transfer matrix type arguments as follows. Let us denote the object on the left hand side by $T(z,p)$ . From its definition one can see that $T(z,p)=[T(1,p)]^z$ and so we can expand $T(z,p)$ in terms of projectors onto the eigenvectors of $T(1,p)$. Equation 2.44 approximates this by retaining only the leading eigenvector, which can be shown to be a product over all circle states. We have added a comment to this effect in the manuscript

Comment 4:
Label under Figure 7. There is probably a typo of $\to$ or. They mention random unitaries, but do they mean their $q\to\infty$ infinity limit?

Our reply:
We have corrected the caption to remove the typo and clarify the meaning. Indeed, we mean the large $q$ limit.

Attachment:

referee_report_1_crosscaps.pdf

---

## Round 1 · Referee Report · Anonymous (Referee 2) · 2025-5-20

Strengths

  1. it performs a systematic study of quantum quenches from initial states with a large amount of long-range entanglement.
  2. both brickwork quantum circuits and Hamiltonian dynamics have been studied; both integrable and chaotic systems have been studied; analytical and numerical results are compared when possible.
  3. it proposes modifications to the quasiparticle and membrance picture that were previously established for studying entanglement dynamics.

Weaknesses

no obvious weakness

Report

The paper by Chalas et al. studies quench dynamics in several cases starting from crosscap states. These states have special structure in that lattice sites that are separated by a long distance are entangled. In most previous studies about quantum quenches, the initial states are chosen to be simple ones with low entanglement (product states or eigenstates of certain Hamiltonians). Going beyond this paradigm can potentially reveal some other interesting phenomena. This paper makes a good step along this direction. The investigations are quite comprehsive and sufficient details are given. Physical implications have been unveiled based on suitable generalization of the quasiparticle and membrane pictures.

It would be helpful to clarify a few minor issues.

  1. It would be good to emphasize that Eq. (1.1) is a rather general definition. It includes not only the crosscap states defined in conformal field theory (CFT), but also many other possibilities that most likely do not correspond to crosscap states in CFT.

  2. It should be mentioned that states like Eq. (1.1) have also been termed "entangled antipodal pair" states in some papers [Phys. Rev. Lett. 133, 170404 (2024); Phys. Rev. Res. 6, L042062 (2024); 2412.18610]

  3. It seems that all calculations are performed using periodic boundary conditions. Is there any special reason? When studying crosscap states in CFT, this is necessary because they are not defined for open boundary conditions. For the current setting, especially considering the possibility of experimental investigations using quantum simulators, open boundary conditions may be useful to explore.

  4. In the case of Hamiltonian dynamics, if one turns to models that cannot be solved exactly by Bethe ansatz, will the quasiparticle and/or membrane pictures survive? The authors may comment on this issue briefly.

Recommendation

Publish (surpasses expectations and criteria for this Journal; among top 10%)

  • validity: top
  • significance: good
  • originality: high
  • clarity: high
  • formatting: excellent
  • grammar: excellent

Author:  Konstantinos Chalas  on 2025-10-14  [id 5934]

(in reply to Report 2 on 2025-05-20)

We would like to thank the referee for carefully considering and evaluating our manuscript.
The questions and remarks were particularly helpful to make the manuscript better and clear.
Moreover, we are particular indebted for suggesting the references.
We proceed by replying to the following questions made by the referee in a point-by-point fashion.

Comments of the referee:
It would be helpful to clarify a few minor issues.

Comments 1-2:
It would be good to emphasize that Eq. (1.1) is a rather general definition. It includes not only the crosscap states defined in conformal field theory (CFT), but also many other possibilities that most likely do not correspond to crosscap states in CFT. It should be mentioned that states like Eq. (1.1) have also been termed "entangled antipodal pair" states in some papers [Phys. Rev. Lett. 133, 170404 (2024); Phys. Rev. Res. 6, L042062 (2024); 2412.18610]

Our reply:
We have added this to the draft and the suggested references.

Comment 3:
It seems that all calculations are performed using periodic boundary conditions. Is there any special reason? When studying crosscap states in CFT, this is necessary because they are not defined for open boundary conditions. For the current setting, especially considering the possibility of experimental investigations using quantum simulators, open boundary conditions may be useful to explore.

Our reply:
We chose to explore only systems with periodic (or anti periodic in the case of fermions) boundary conditions as it was the most straightforward to analyse using the techniques that we have used. As the referee points out, it would be possible to examine systems with different boundary condition. Given the amount of material present in the current draft, we will leave examination of this interesting point to future work.

Comment 4:
In the case of Hamiltonian dynamics, if one turns to models that cannot be solved exactly by Bethe ansatz, will the quasiparticle and/or membrane pictures survive? The authors may comment on this issue briefly.

Our reply:
This is an interesting question and worthy of further consideration. Typically, in the presence of weak integrability breaking one might expect features of the quasi-particle picture to persist at least on time scales shorter than the quasiparticle lifetime. In the present case, however, features of the quasiparticle picture appear on time scales which scale with system size. The competition between the two effects would be interesting to explore. We have added a comment in the conclusions to reflect this.

Attachment:

referee_report_2_crosscaps_.pdf

---

## Round 1 · Referee Report · Anonymous (Referee 3) · 2025-5-31

Strengths

1- Investigates a new type of quench 2- Examines multiple cases (quantum circuits and spin chains, both ergodic and integrable) 3- Includes both analytical and numerical analyses 4- Provides physical interpretations 5- Well-written

Weaknesses

The models under investigation are introduced rather briefly.

Report

The article investigates quantum quenches of crosscap states. Unlike the initial states typically considered in the literature, which exhibit short range entanglement, crosscap states are characterized by long-range entanglement. The authors explore crosscap quenches in both quantum circuits and spin chains, employing analytical and numerical methods. The primary quantities analyzed are entanglement entropy and mutual information. The results are accompanied by physical interpretations and are found to be consistent with both quasiparticle and membrane-based descriptions.

Based on the specific models studied, the authors observe that in chaotic systems, entanglement entropy remains constant, whereas in integrable systems, it initially stays constant but eventually begins to decrease. For chaotic systems, mutual information decreases until it vanishes and remains zero, while in integrable systems, it decreases initially but later increases again.

Requested changes

The article contains several points that could benefit from clarification:

1) The discussion of Random Unitary Circuits could be elaborated further, particularly regarding the origin of equation (2.34).

2) What is meant by "a specific class of dual unitary gates" prior to equation (2.44)? Does this refer to all chaotic gates? Additionally, what does "for large enough z" signify in equation (2.44)?

3) In equation (3.8), why is N_A conserved? The commutation [N,H]=0 alone is insufficient; some condition on the initial state must also be specified.

4) There appears to be a parenthesis mismatch in equation (3.9).

5) In the case of the XXZ spin chain, the description of bound states and the TBA are valid in the infinite L limit. However, the crosscap state is ill-defined in this limit. Does this not lead to a contradiction?

Recommendation

Publish (easily meets expectations and criteria for this Journal; among top 50%)

  • validity: high
  • significance: good
  • originality: high
  • clarity: high
  • formatting: perfect
  • grammar: -

Author:  Konstantinos Chalas  on 2025-10-14  [id 5935]

(in reply to Report 3 on 2025-05-31)
Category:
answer to question

We would like to thank the referee for carefully reading and reviewing our manuscript. The questions are indeed particularly useful to clarify certain subtle points regarding the analysis.

Comment 1:
The discussion of Random Unitary Circuits could be elaborated further, particularly regarding the origin of equation (2.34).

Our reply:
We have added a comment regarding this below the equation.

Comment 2:
What is meant by "a specific class of dual unitary gates" prior to equation (2.44)? Does this refer to all chaotic gates? Additionally, what does "for large enough z" signify in equation (2.44)?

Our reply:
This sentence is unintentionally vague and we have amended these statements. Dual unitary circuits have only been completely classified for $q=2$. This parametrization can be extended to arbitrary values of $q$, however, for $q>3$ there exist other classes of dual unitary gates. Equation (2.44) has been proven for this, special class of dual unitary circuits which encompasses all $q=2$ cases. The proof essentially entails showing that the transfer matrix constructed from the dual unitary circuits has a unique largest eigenvector with eigenvalue $1$. So, when it is taken to a large power it becomes a projector onto this leading eigenvalue as is the case on the right hand side. This drops terms which are of the order of the $\lambda^z$, $\lambda$ is the next largest eigenvalue. We have added some additonal remarks on this to the text.

Comment 3: In equation (3.8), why is $N_A$ conserved? The commutation $[N,H]=0$ alone is insufficient; some condition on the initial state must also be specified.

Our reply:
The subsystem fermion number, $N_A$ is not conserved. However, the expectation value, $\langle N_A(t)\rangle$ remains constant as a consequence of the translation invariance of the system.

Comment 4:
There appears to be a parenthesis mismatch in equation (3.9).

Our reply:
This has been corrected.

Comment 5:
In the case of the XXZ spin chain, the description of bound states and the TBA are valid in the infinite L limit. However, the crosscap state is ill-defined in this limit. Does this not lead to a contradiction?

Our reply:
It is correct that the string structure of the Bethe roots is only valid in the thermodynamic limit. However, it has now been well established that the quasiparticle picture can be used in TBA integrable models, wherein the different string types correspond to different quasiparticle species. Moreover it is known that the quasiparticle picture compares excellently to exact numerics even for finite systems, both free and interacting. We have studied free fermions exactly in our scenario and shown that a quasiparticle picture emerges in that case despite the finiteness of the system. This provides the justification for the use of the quasiparticle picture in the interacting case also.

Attachment:

referee_report_3_crosscaps.pdf

---

## Round 2 · Author Response

The manuscript is resubmitted with the changes requested by the referees.

---

## Round 2 · List of Changes

• Removed an unfinished sentence on page 6, second last paragraph
• simplified equation (3.20)
• several typos and grammatical errors have been fixed
• highlighted the need for anti periodic boundary conditions below equation 3.3
• modified the discussion of the entanglement membrane picture in section 2.2
• added an explanation of equation 2.44
• added a comment and references below equation 1.1. regarding EAP states.
• Added a comment regarding nonintegrable models to the conclusion

---

## Editorial Decision

published